# Navigating the credibility risks of environmental scientists' activism
J. Lukas Thürmer [1,2,3] ✉, Jeremias Braid [1], Sean M. McCrea [4] & Matthew J. Hornsey [5]

Cost-benefit analyses of whether environmental scientists should engage in activism currently rest on a thin empirical base, despite a lively debate on the topic. There are several potential benefits of scientists' activism, but some have argued that these benefits might be offset by the potential for activism to undermine public perception of environmental scientists as unbiased and competent. To explore these potential consequences, we asked participants to read two (ostensibly real) profiles of climate scientists that either described themselves as activists or not. Study 1 ($N = 491$) found that a scientist who engaged in conventional activism was seen as slightly less competent and more hypocritical than a scientist who engaged in public science communication, but there was no impact on their persuasiveness. Study 2 ($N = 636$, pre-registered) found that a scientist who engaged in civil disobedience, a more disruptive form of activism, was seen as less competent and more hypocritical than a non-activist scientist who only engaged in teaching and research, with predicted spill-over effects on trust in the scientist's field. Scientist activists were also downgraded on a range of other dimensions. We conclude that engaging in activism has small but reliable costs for environmental scientists.

There is a long-standing debate within the scientific community about whether environmental scientists should engage in activism. On one side are those who believe it is environmental scientists' responsibility to engage in activism, ranging from conventional protests to civil disobedience[1]. Several arguments have been proposed in favour of this notion: that traditional influence strategies are not working[2], that activism has documented success as an influence strategy[3,4], that science cannot (and must not) be value-free[5], that scientific activism is a signal of the urgency of the environmental problem[6], that scientists' voices have especially strong influence on the public, policymakers and other activists[7], and that there is an intrinsic moral or ethical imperative to use that influence[8,9]. According to this view, activism is an effective tool to increase the salience of the environment as an important issue, to show one's conviction, and to sway public opinion.

On the other side of the debate are those who are concerned that activism threatens the credibility of scientists as unbiased, honest brokers of information. From this perspective, it is scientists' reputation as impartial observers that makes scientific voices uniquely influential, and if activism threatens that reputation, the long-term cost outweighs the benefits[10–12]. Concerns about the reputational cost of activism are reinforced by research showing that the general population associates activists with negative stereotypes such as being militant, self-righteous, and ideologically biased[13,14].

Surveys of scientists themselves suggest generally positive views of environmental activism, although approximately 10% expressed concerns that participating in protests might create tensions with their role as a scientist (e.g., reduce their perceived objectivity)[15]. An international survey of climate scientists also revealed a generally harmonious perception of the relationship between science and activism, although this perception was most pronounced among the roughly half of the sample who were already members of an activist group such as Extinction Rebellion or Greenpeace[16]. Members of activist groups were also less uncertain about the effectiveness of activism.

In weighing up the potential credibility costs of engaging in activism, environmental scientists have a surprisingly thin evidence base to draw on. In evaluating the evidence, it is important to distinguish between advocacy and activism. Advocacy typically refers to activities in which scientists use their expertise to communicate, recommend, or support evidence-based solutions to policymakers or the public, usually within conventional institutional channels. Activism, in contrast, typically refers to direct public action—such as protests, demonstrations, or campaigns—that explicitly calls for social or political change and may be perceived as more confrontational. Although both are forms of public engagement, activism carries a stronger association with partisanship and moral urgency, which may

[1]Department of Psychology, University of Salzburg, Salzburg, Austria. [2]Economic Psychology, Seeburg Castle University, Seekirchen am Wallersee, Austria. [3]Salzburg Center for European Union Studies, University of Salzburg, Salzburg, Austria. [4]Department of Psychology, University of Wyoming, Laramie, WY, USA. [5]Business School, The University of Queensland, St Lucia, QLD, Australia. ✉e-mail: lukas.thuermer@plus.ac.at

make it more contested in terms of public perceptions of scientific credibility.

A host of research has focused on advocacy (taking a public stance in support of a cause) rather than activism as traditionally interpreted (e.g., petitioning, marching, civil disobedience). The most commonly cited paper reported that advocacy (operationalised as an interview with Associated Press) did not have consistently negative effects on scientists' perceived credibility, and none at all on trust in the broader climate science community[17]. A further experiment found that non-controversial advocacy (operationalised as writing Op-Eds presenting information and suggesting education campaigns) increased climate scientists' credibility compared to only providing scientific information, although this benefit disappeared when the advocacy included recommendations of regulation and mandating of solutions [18].

Non-experimental work on public perceptions has typically comprised surveys or focus groups. Surveys in the U.S. and Germany indicated that the general public is less supportive than is the scientific community of scientists' political and public engagement[19]. Furthermore, qualitative analyses of 15 focus groups of citizens, journalists, and scientists in Denmark indicated high levels of support for the notion that scientists ought to engage in neutral rather than politically prescriptive communication[11]. A recent survey corroborates this view, as 4 out of 10 Americans indicated they expected scientists to focus on establishing sound scientific facts rather than contributing to policy debates[20]. Others have observed that a slight majority of the overall U.S. population (51%) agrees that scientists should take an active role in public policy debates, although this agreement was substantially higher among Democrats (67%) than Republicans (35%)[21]. Globally, support for scientists' advocacy was observed at a comparable level (54%), with only 20% disagreeing or strongly disagreeing[22]. Overall, these studies suggest generally neutral-to-positive views about scientists' advocacy, although this support does not appear to be universal.

A smaller set of studies has investigated the influence of scientists' activism. A survey conducted immediately before and after the 2017 March for Science rallies in the U.S. revealed a polarisation effect: Liberals' attitudes toward scientists became more positive, whereas conservatives' attitudes became more negative[23]. A recent, high-powered registered report used vignettes to investigate the influence of scientists' participation in protests against oil and gas drilling in an experimental design. Vignettes either did not mention scientists (control condition), mentioned a scientist's endorsement of protests, or mentioned a scientist's active participation. The study found no evidence that scientists' participation in activism undermined their credibility or public trust in environmental science as compared to only endorsing the protests (this variable was not assessed in the control condition). However, it also found little evidence that scientists' endorsement of or engagement in activism increased public support for related climate policies, activist movements, or donations to environmental causes[24]. Crucially, no non-activist scientist condition was included in the study, and it thus cannot directly evaluate the pros and cons of scientists engaging in activism. Finally, a recent experiment observed that college students indicated greater personal worry about climate change after reading a mock newspaper article by a scientist engaging in activism rather than a scientist just teaching[25]. Indicating that the scientist engaged in civil disobedience (i.e., was arrested for protesting) further increased reported personal worry about climate change. This study did not observe negative effects on scientists' apparent competence or credibility. However, the presented information source was highly trustworthy (a newspaper article), and the sample of college students was left-leaning and highly supportive of environmental action.

To provide direct causal evidence for the benefits or costs of scientists' environmental activism, the current studies involved demographically diverse Americans reading two (ostensibly real) profiles of environmental scientists. Study 1 compared a scientist who engaged in conventional environmental activism with a scientist who engaged in public science communication but with no reference to activism. Scientists were presented with carefully controlled profile pictures merged onto AI-generated background scenes in a city, either including environmental protesters or not. Study 2 compared an environmental scientist who engaged in civil disobedience, a more disruptive form of activism[5,6,25], with a scientist who engaged in environmental research and teaching activities. Moreover, the civil disobedience manipulation and measures were adapted from past work[25]. As such, both studies controlled for whether the target engaged in environmental research, manipulating only whether the target engaged in activist behaviour as part of their engagement. Potential confounding factors such as target sex, matching of the materials to conditions, and the presentation order of materials were fully randomised (counter-balanced). After each comment, participants evaluated the scientist (Studies 1 and 2) and their research field (Study 2). We compared the mean evaluations of scientist activists versus scientist non-activists to test the causal effect of scientists' activism on evaluations.

To summarize, we explored (Study 1) and tested (Study 2) the following hypotheses:

H1: Environmental scientists engaging in non-disruptive activism (vs. public science communication; Study 1) or disruptive activism (vs. not engaging in activism at all; Study 2) will damage their scientific reputation.

H1a: Environmental scientist-activists will be evaluated as less competent (lower expertise).

H1b: Environmental scientist-activists' findings will be evaluated as less credible.

H2: Environmental scientists engaging in disruptive activism (vs. not engaging in activism) will be evaluated to be more hypocritical.

H3: Reading a description of environmental scientists engaging in disruptive activism (vs. not engaging in activism) will lead to lower trustworthiness evaluations of their research field (Study 2 only).

Both studies also included exploratory moderators of these effects, based on our intuition and past research. Specifically, we measured past environmental behaviour as people who already engage in pro-environmental behaviours may be more sympathetic to environmental scientists' activism than less engaged individuals. Similarly, we included measures of trust in science, as participants with higher trust in science may be more likely to extend this trust to scientists engaging in activism as compared to less trusting individuals. Conversely, we included conspiracy mentality as a higher conspiracy mentality could intensify scepticism toward scientists when they are seen as political actors. We also included trait-level reactance and resistance to change as exploratory moderators, reflecting the idea that responses to scientist activism may depend on people's general motivation to protect autonomy and stability. Those high in reactance may view activist scientists as overstepping or coercive, triggering defensive scepticism, whereas those high in resistance to change may oppose activism because it symbolises disruption to the social or political status quo.

## Methods

### Transparency, inclusion, and ethics

We report all measures, manipulations, and participant exclusions. Materials, analyses, and data (Studies 1 and 2) are available at https://osf.io/pshak. Study 2 was pre-registered on September 19, 2025 before launching the data collection at https://osf.io/2ujrg. Unless otherwise noted, our analyses followed the pre-registered procedures. The University of Salzburg internal review board provided ethics approval for the studies (GZ10/2020). All researchers involved, including a PhD candidate, are listed as authors on this paper.

### Participants and design

In Study 1, we used CloudResearch[26] to recruit Amazon's Mechanical Turk workers as well as Prime-Panels participants located in the U.S., with >85% approval rate and at least 100 completed HITs. CloudResearch was set up to block suspicious geocode locations, workers on its universal exclude list, and duplicate IP addresses, as well as to verify worker country and state location, and micro-batch our HIT. Workers received a base-flat-payment of $1USD. On July 22, 2024, 554 participants completed the study. Our raw dataset

## Table 1 | Sample Demographics

| Measure | Study 1 | Study 2 |
|---|---|---|
| Gender | 245 male, 238 female, 2 non-binary, and 1 diverse | 310 male, 309 female, 14 non-binary, and 1 diverse |
| Age | $M = 47.94$ years ($SD = 16.71$) | $M = 46.77$ years ($SD = 16.06$) |
| Race | 376 White, 69 African American, 10 Latino, 17 Asian, and 14 other | 479 White (Non-Hispanic), 71 Black or African American, 28 Hispanic or Latino/a/x, 27 Asian or Asian American, 19 participants either chose more than one option or wrote "Mixed Race", 4 Native American or Alaska Native, 3 other (not specified), 2 Middle-Eastern/North African, 1 Asian Caucasian Mix |
| Self-description as activist/scientist | Activist $M = 3.02$ ($SD = 2.08$) Scientist $M = 2.26$ ($SD = 1.88$) | Activist $M = 2.28$ ($SD = 1.65$) Scientist $M = 1.80$ ($SD = 1.52$) |
| Educational attainment | 14 did not complete high school, 8 General Educational Development Test (i.e., college entry test), 124 high school diplomas, 44 postsecondary vocational certificates, 65 associate's degrees, 156 bachelor's degrees, 70 master's degrees, and 5 doctoral degrees (Ph.D., JD., MD) | 7 did not complete high school, 16 General Educational Development Test (i.e., college entry test), 171 high school diploma, 36 postsecondary vocational certificate, 88 associate's degree, 195 bachelor's degree, 92 master's degree, and 20 doctoral degree (Ph.D., JD., MD) |
| Income | 26 Less than $10,000, 29 $10,001 to $15,000, 55 $15,001 to $25,000, 137 $25,001 to $50,000, 120 $50,001 to $75,000, 61 $75,001 to $100,000, 38 $100,001 to $150,000, and 20 more than $150,000 | 16 Less than $10,000, 20 $10,001 to $15,000, 51 $15,001 to $25,000, 127 $25,001 to $50,000, 152 $50,001 to $75,000, 122 $75,001 to $100,000, 101 $100,001 to $150,000, and 42 more than $150,000 |
| Political orientation | n/a | $M = 4.72$ ($SD = 3.16$) |
| SES ladder | n/a | $M = 4.96$ ($SD = 1.70$) |

includes 222 incomplete entries and test-runs which are not included in this number; analyses did not show systematic attrition (see Supplementary for details). Two participants reported technical issues and 61 failed the attention check (see below) leaving $N = 491$ for analyses (sample demographics in Table 1). Sensitivity analyses[27] indicated that this sample was sufficient to detect a small effect of $d = 0.16$ at $1 − \beta = 0.95$ or a negligible effect of $d = 0.13$ at $1 − \beta = 0.80$. The sample was also sufficient to explore moderators yielding a small-to-medium interaction effect of $f = 0.16$ at $1 − \beta = 0.95$ or a small effect of $f = 0.13$ at $1 − \beta = 0.80$.

To provide a confirmatory test of our hypotheses, Study 2 was pre-registered and recruited a larger sample, providing a high power to detect a small-to-medium effect. A power analysis using G*power 3.1 (Faul, Erdfelder, Buchner, & Lang, 2009), assuming the effect observed in Study 1 ($d = 0.15$) and setting $1 − \beta = 0.95$, resulted in a minimum sample size of $N = 580$ for a paired-samples $t$-test (two-tailed). We aimed to recruit 650 subjects to account for potential drop-outs (see below). The sample was also sufficient to explore moderators yielding a small-to-medium interaction effect of $f = 0.14$ at $1 − \beta = 0.95$ or a small interaction effect of $f = 0.11$ at $1 − \beta = 0.80$. We used Prolific Academic to recruit a nationally representative sample (stratified by age, gender, and political affiliation) of participants located in the U.S. Participants received a base-flat-payment of the equivalent of £1.50 (about $2USD). On September 19, 2025, 651 participants completed the study. Our raw dataset includes 129 incomplete entries, which are not included in this number; analyses did not show systematic attrition (see Supplementary for details). Among the complete data sets, 15 participants failed the attention check (see below), leaving $N = 636$ for analyses (demographics in Table 1).

Both experiments followed 2-level (scientist activist vs. scientist non-activist) within-participants designs. Order of vignettes, order of conditions, matching of faces to conditions (Study 1), the assignment of scientists' names to the vignettes (Study 2), and the matching of the scientists' research field to the vignettes (Study 2) were fully counterbalanced (i.e., randomised). To control for target sex, participants either viewed two male or two female targets (i.e., target sex was randomised between participants), either by showing respective photos (Study 1) or by varying the scientist's name (Study 2). These variations were applied automatically by the employed survey software formR [28].

### Procedure

Participants responded to the study advertisements on Prime Panels or MTurk (Study 1) or Prolific Academic (Study 2) by clicking on a link to a web-based survey in formR[28]. In both studies, participants provided informed consent and, after the main part of the study, were fully debriefed regarding origin of the comment at the end of the respective survey. All participants were given the option to withdraw their consent, which none of them did. The scales in both studies contained an attention check asking participants to select a specific response (Select *strongly agree* so that we are sure you are reading attentively).

**Study 1.** The main part of Study 1 proceeded as follows: After providing informed consent, participants were asked to consecutively read two statements, ostensibly written by the target. Target person images were selected from the Chicago Face Database[29], a set of standardised and rated portrait pictures. Pictures were selected to reduce the variance in the stimulus materials; that is, all were Caucasian men and women, matched for average attractiveness, trustworthiness, and prototypicality, with a rated age range of 32.88 to 41.69 years. Four sets of faces were chosen (Chicago Face Database items CFD-WF-217−085-N, CFD-WF-245−084-N, CFD-WM−018−002-N, CFD-WM-248−036-N).

We generated raw scenes of climate activist demonstrations in a city (Fig. 1) using the AI tool Microsoft Designer/Copilot (model: DALL-E 3) and replaced the faces in the generated scene with the selected Chicago Face Database portraits using Adobe Photoshop. For the scientist non-activist condition, we instructed Adobe Photoshop's Generative Fill to adapt the generated scenes so that no climate demonstration took place (i.e., we removed all people except the target person and activism materials such as 'save the planet' signs). The comments all indicated that the target was an environmental scientist working on a topic related to climate change (i.e., energy use or waste reduction). In the scientist activist condition, the statement indicated that they like to attend rallies and write to politicians to influence the public. In the scientist-only condition, the statement indicated that they frequently share their findings with the media and in public lectures (Table 2).

After reading about each target, participants then completed measures of nine dependent variables, details of which can be found in Table 3. The first five measures were designed to assess the target's characteristics, with two items each measuring facets of friendliness, morality, competence, and assertiveness[30], and an 8-item scale measuring the target's perceived hypocrisy[31]. The person evaluation scale covered the most common dimensions of social evaluation[30]; we additionally included hypocrisy as it represents a common response to social motive violations and is related to message rejection[31]. Participants then rated the extent to which the message was perceived to be constructive (1 item) and participants' attribution of environmental concern to the message source (2 items)[32,33] as well as their

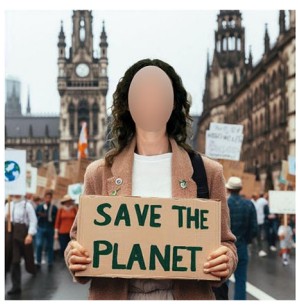
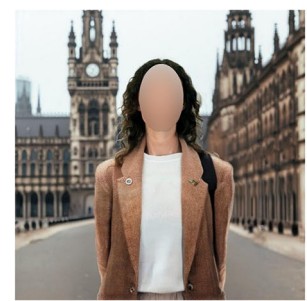
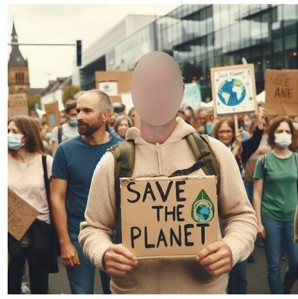
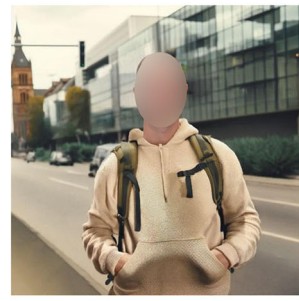
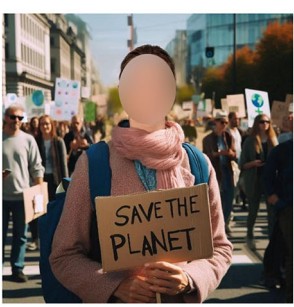
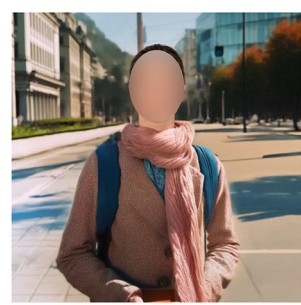
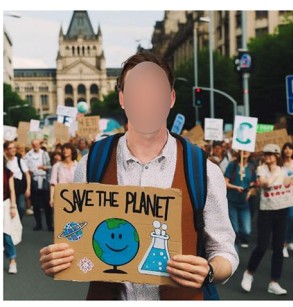
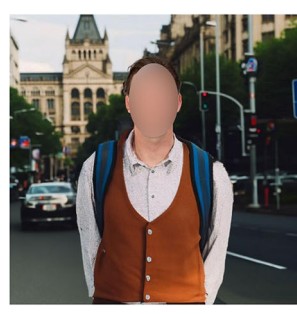

**Fig. 1 | Stimulus pictures of scientist activists (left panel) and scientist non-activists (right panel).** Chicago Face Database photos were used (CFD-WF-217−085-N, CFD-WF-245−084-N, CFD-WM−018−002-N, CFD-WM-248−036-N); since no personal consent for the publication of these images could be obtained from the photographed people, faces are blurred here.

agreement with the target's message (2 items) and issue salience (3 items)[34]. Finally, for exploratory purposes, we incorporated three open-ended questions (What do you think about the issues addressed in the comment?, What do you think about the person who made the comment?, What do you think are the reasons why this person made that comment?).

After evaluating the second comment, participants completed additional measures to assess potential moderators. As detailed in Table 3, moderators were reactance (how much regulation attempts and rules typically instil countering behaviour in participants), pro-environmental behaviour (engagement in more environmentally friendly behaviour such as biking instead of driving), conspiracy mentality (the endorsement of the idea that secret powers control world affairs), trust in science (participants' general trust that scientists and the scientific method yield reliable results), and resistance to change (8 items).

Finally, participants provided demographic information (age, gender, race, educational attainment, income) and indicated whether they were activists/scientists themselves (How much would you describe yourself as an activist?; If so, in which areas? How much would you describe yourself as a scientist?; If so, in which areas?). Age, participant gender, and target sex were included in the suite of moderation analyses.

**Study 2.** The main part of Study 2 proceeded as follows: After providing informed consent, participants were first asked to respond to scales and questions designed to assess demographics and potential moderators. We moved the moderators to the beginning of the survey to preclude an effect of reading the scientist vignettes on these measurements. Specifically, participants provided demographic information (age, gender, race, educational attainment, income) and indicated whether they were activists/scientists themselves (How much would you describe yourself as an activist?; If so, in which areas?; How much would you describe yourself as a scientist?; If so, in which areas?). We then assessed the moderators observed in Study 1, pro-environmental behaviour, trust in science, and conspiracy mentality (Table 3). The measures on reactance and resistance to change were not included because they did not moderate any of the effects observed in Study 1.

Participants then consecutively read two descriptions of ostensibly real scientists (Table 4). Study 2 used text-only materials as the AI-generated background-scenes used in Study 1 may have been unconvincing for some participants. Both vignettes briefly described a scientist working on environmental issues (consumer products or energy consumption). The rest of the paragraph was designed to manipulate the scientists' engagement in activism. The scientist activism condition was directly adapted from Friedman (2024)[25] and described the targets' involvement in civil disobedience (e.g., being arrested for blocking train lines to stop the transportation of pollutants), a disruptive form of activism. In the scientist non-activism condition, the scientist was described as engaging in climate-related teaching and research activities (e.g., organizing an environmental science lecture series).

After reading about each target, participants indicated the target's name and research topic (as comprehension checks) and then completed measures of eight dependent variables, details of which can be found in Table 3. The first three measures were designed to assess trust in the target, with six items measuring expertise-based trust, four items measuring integrity-based trust, and four items measuring benevolence-based trust[35]. Expertise-based trust served as our registered dependent variable to test Hypothesis 1a. The next three items were directly adapted from Friedman (2024)[25]. Two items were designed to assess participants' trust in the scientists' findings and one to assess participants' intention to act on them. The first measure served as our dependent variable to test Hypothesis 1b. The 8-item hypocrisy scale[31] (DV to test H2) and two items on the attribution of environmental concern to the scientist[32,33] used in Study 1 followed. A final measure of general trust in the scientists' research field (three items) served as the dependent measure of Hypothesis 3. The measures on message constructiveness, agreement with the target's message, issue salience, and the free text items were not included because they did not yield useful information in Study 1.

## Analyses

We used R[36], including multiple packages to analyze our data (see Supplementary for details). All scale reliabilities were satisfactory, and we thus computed single indexes by averaging the items (Table 3). Since our study followed a within-subjects design, checks for variance homogeneity were not necessary. Data distribution was assumed to be normal, but this was not formally tested. We assumed normality of differences and residuals due to the large sample sizes (Study 1 $N = 491$; Study 2 $N = 636$).

## Results
### Descriptive analyses and counter-balancing checks
Condition-independent means (across 982 data points, Study 1, and 1,272 data points, Study 2) for the dependent variables are displayed in Table 3. Evaluations of scientists were generally positive, as indicated by relatively

**Table 2 | Stimulus texts by scientist activist vs. scientist non-activist condition and message content counterbalancing (Study 1)**

| Condition and stimulus text | Version 1 | Version 2 |
|---|---|---|
| Activist condition, waste reduction | "I've been working as a researcher in environmental science for over a decade. In my research, I investigate how people's choices affect our climate and nature. I study why some people decide to recycle their waste, while others don't. Understanding this in more detail helps us find better ways to encourage eco-friendly habits that can slow down global warming and protect the environment. I hope that my research encourages people to recycle their waste. I also consider myself a climate activist and I'm part of a couple of environmental activist groups. I don't just stay in the lab, I like to get out there and attend rallies, write to parliamentarians, influence the general public and generally try to make a difference when it comes to climate change. My research is closely aligned to my personal passions and my politics and I don't see a strong divide between my research world and my activist world." | "I've spent most of my life studying and researching issues to do with environmental sustainability. My research delves into the influence of individual decisions on environmental health. I explore the motivations behind why certain individuals prioritize recycling, while others appear indifferent. Gaining a deeper insight into these behaviours is crucial for developing strategies that promote green practices, potentially decelerating the pace of climate change. In this way, my research can help people recycle their waste. I also consider myself a climate activist and I'm part of a couple of environmental activist groups. I don't just stay in the lab, I like to get out there and attend rallies, write to parliamentarians, influence the general public and generally try to make a difference when it comes to climate change. My research is closely aligned to my personal passions and my politics and I don't see a strong divide between my research world and my activist world." |
| Activist condition, saving energy | "I've been working as a researcher in environmental science for over a decade. In my research, I investigate how people's choices affect our climate and nature. I study why some people decide to use less energy, while others don't. Understanding this in more detail helps us find better ways to encourage eco-friendly habits that can slow down global warming and protect the environment. I hope that my research encourages people to use less energy. I also consider myself a climate activist and I'm part of a couple of environmental activist groups. I don't just stay in the lab, I like to get out there and attend rallies, write to parliamentarians, influence the general public and generally try to make a difference when it comes to climate change. My research is closely aligned to my personal passions and my politics and I don't see a strong divide between my research world and my activist world." | "I've spent most of my life studying and researching issues to do with environmental sustainability. My research delves into the influence of individual decisions on environmental health. I explore the motivations behind why certain individuals save energy, while others appear indifferent. Gaining a deeper insight into these behaviours is crucial for developing strategies that promote green practices, potentially decelerating the pace of climate change. In this way, my research can help people save more energy. I also consider myself a climate activist and I'm part of a couple of environmental activist groups. I don't just stay in the lab, I like to get out there and attend rallies, write to parliamentarians, influence the general public and generally try to make a difference when it comes to climate change. My research is closely aligned to my personal passions and my politics and I don't see a strong divide between my research world and my activist world." |
| Non-activist condition, waste reduction | "I've been working as a researcher in environmental science for over a decade. In my research, I investigate how people's choices affect our climate and nature. I study why some people decide to recycle their waste, while others don't. Understanding this in more detail helps us find better ways to encourage eco-friendly habits that can slow down global warming and protect the environment. I hope that my research encourages people to recycle their waste. I attend conferences and network with other scientists because I feel passionately about my research area and want to spread the word. I think it's important that the research I do and the research my colleagues do is broadcast in non-scientific circles as well. I convey my findings as a neutral expert. I like to write science pieces for the general public, talk to newspapers, and give TV interviews so they can understand what is going on in our research." | "I've spent most of my life studying and researching issues to do with environmental sustainability. My research delves into the influence of individual decisions on environmental health. I explore the motivations behind why certain individuals prioritize recycling, while others appear indifferent. Gaining a deeper insight into these behaviours is crucial for developing strategies that promote green practices, potentially decelerating the pace of climate change. In this way, my research can help people recycle their waste. I attend conferences and network with other scientists because I feel passionately about my research area and want to spread the word. I think it's important that the research I do and the research my colleagues do is broadcast in non-scientific circles as well. I convey my findings as a neutral expert. I like to write science pieces for the general public, talk to newspapers, and give TV interviews so they can understand what is going on in our research." |
| Non-activist condition, saving energy | "I've been working as a researcher in environmental science for over a decade. In my research, I investigate how people's choices affect our climate and nature. I study why some people decide to use less energy, while others don't. Understanding this in more detail helps us find better ways to encourage eco-friendly habits that can slow down global warming and protect the environment. I hope that my research encourages people to use less energy. I attend conferences and network with other scientists because I feel passionately about my research area and want to spread the word. I think it's important that the research I do and the research my colleagues do is broadcast in non-scientific circles as well. I convey my findings as a neutral expert. I like to write science pieces for the general public, talk to newspapers, and give TV interviews so they can understand what is going on in our research." | "I've spent most of my life studying and researching issues to do with environmental sustainability. My research delves into the influence of individual decisions on environmental health. I explore the motivations behind why certain individuals save energy, while others appear indifferent. Gaining a deeper insight into these behaviours is crucial for developing strategies that promote green practices, potentially decelerating the pace of climate change. In this way, my research can help people save more energy. I attend conferences and network with other scientists because I feel passionately about my research area and want to spread the word. I think it's important that the research I do and the research my colleagues do is broadcast in non-scientific circles as well. I convey my findings as a neutral expert. I like to write science pieces for the general public, talk to newspapers, and give TV interviews so they can understand what is going on in our research." |

Each participant read two texts, one in each type and one in each condition. The assignment of the type, the order, and the content was fully randomized.

high ratings of competence, high attributions of concern for the environment, and low hypocrisy scores (Study 1), as well as trait ratings and attributions of concern for the environment above the mid-point (Study 2) across the conditions. We observed no significant differences across Topics (Energy saving vs. Recycling [Study 1] or Energy consumption vs. Consumer products [Study 2]) and no significant Condition × Completion Time interactions (see Supplementary for details).

### Manipulation effects
Comparisons of means across conditions are summarised in Table 5 (Study 1) and Table 6 (Study 2). The tables also include the results of the paired samples *t*-tests used to test our hypotheses, as we pre-registered for Study 2.

**Study 1**. Activist scientists were evaluated as significantly less competent than non-activist scientists, and also as more hypocritical. These effects were small but reliable after a Bonferroni adjustment for 9 repeated significance tests. Activists' comments were also evaluated to be less constructive than non-activists' comments, although this effect was no longer significant after Bonferroni correction. We observed no effects of scientists' activism on their perceived morality or friendliness. Furthermore, there were no significant effects of condition on the attribution of environmental concern, issue salience, or agreement with the comment. Thus, while we observed negative consequences of scientists' activism on person evaluation, no significant effects on message perception emerged.

**Table 3 | Scales, item wording, and sources with grand means and standard deviations (Study 1 and Study 2)**

| Concept | Wording | Source | Study 1 M (SD) | Study 2 M (SD) |
|---|---|---|---|---|
| Traits (two items each): Friendliness (F) (S1 αs = 0.89; 0.91) Morality (M) (S1 αs = 0.89; 0.89) Competence (C) (S1 αs = 0.90; 0.92) Assertiveness (A) (S1 αs = 0.88; 0.88) | Please rate the extent to which each trait describes the person displayed above: *1 = not at all, 7 = extremely*. …warm (F) …friendly (F) …honest (M) …sincere (M) …capable (C) …skilled (C) …confident (A) …determined (A) | Koch et al. (2024)[30] | 4.91 (1.58) 5.54 (1.46) 5.55 (1.45) 5.73 (1.38) | n/a |
| Hypocrisy (S1 αs = 0.96; 0.98; S2 αs = 0.96; 0.96) | To what extent do you think the person who wrote the comment is…? *1 = not at all, 7 = very much* …a hypocrite. …inconsistent. …pretending to be something they are not. …blaming (or criticizing) others when they do things just as bad or worse. …choosing some values to uphold while being lazy about upholding other values. …failing to practice the very same thing they preach. | Pillow et al. (2023)[31] | 2.86 (1.85) | 2.12 (1.42) |
| Constructiveness (single-item measure) | To what extent do you think the comments were constructive? *1 = not at all, 7 = very much* | Hornsey & Imani (2004)[32] | 5.20 (1.57) | n/a |
| Attribution of environmental concern (S1 αs = 0.82; 0.87; S2 αs = 0.85; 0.86) | To what extent do you think… *1 = not at all, 7 = very much* …the person who wrote these comments cares about the environment? …the comments were made in the environment's best interests? Study 2: …[name of researcher] cares about the environment? …[name of researcher] acts in the environment's best interests? | Hornsey & Imani (2004)[32] | 5.69 (1.36) | 6.08 (1.25) |
| Agreement with the comment (S1 αs = 0.78; 0.77) | Please tell us whether you think it can be justified to make a comment as the one you just read. *1 = never justifiable, 7 = always justifiable* How much do you agree with the comment? *1 = not at all, 7 = very much* | McCrea et al. (2022)[44] | 5.39 (1.51) | n/a |
| Issue salience (S1 αs = 0.89; 0.91) | From your perspective, how important is it that the issues addressed in the comment…? *1 = not at all, 7 = very much* …will be discussed further. …are taken seriously by society. …are finally being tackled. | Halfmann et al. (in prep.)[34] | 5.34 (1.50) | n/a |
| Trust: Expertise (E) (7 items) (S2 αs = 0.96; 0.96) Integrity (I) (5 items) (S2 αs = 0.95; 0.96) Benevolence (B) (4 items) (S2 αs = 0.96; 0.97) | Now, please evaluate [name of researcher] on the following scales: *1–7 scales with adjectives as endpoints* competent–incompetent (E) intelligent–unintelligent (E) well educated–poorly educated (E) professional–unprofessional (E) experienced–inexperienced (E) qualified–unqualified (E) helpful–hindering (E) sincere–insincere (I) honest–dishonest (I) just–unjust (I) unselfish–selfish (I) fair–unfair (I) moral–immoral (B) ethical–unethical (B) responsible–irresponsible (B) considerate–inconsiderate (B) | Hendriks et al. (2015)[35] | n/a | 5.83 (1.33) 5.65 (1.38) 5.57 (1.53) |
| Credibility (S2 αs = 0.98; 0.97) | How trustworthy do you believe Dr. [last name of researcher]'s conclusions are likely to be? *0 = Not at all trustworthy, 10 = Extremely trustworthy* How much confidence do you have in Dr. [last name of researcher]'s research findings? *0 = No confidence at all, 10 = Very high confidence* | Friedman (2024)[25] | n/a | 7.43 (2.64) |
| Action intent (single-item measure) | To what extent do you intend to take action to [buying more sustainable products OR help reduce your energy consumption]? *0 = Not at all, 10 = Very much* | Friedman (2024)[25] | n/a | 6.65 (2.91) |
| Trust in the researchers' field (S2 αs = 0.93; 0.93) | More generally, how much do you trust scientists working on the impact of [consumer products OR energy consumption] on greenhouse gas emissions to do each of the following? *1 = Not at all, 7 = A lot* To do their work with the intention of benefiting the public. To be open and honest about who is paying for their work. To communicate their results openly and in an unbiased manner to the public. | Adapted from Sturgis et al. (2021)[45] | n/a | 5.32 (1.54) |
| Open-ended questions | What do you think about the issues addressed in the comment? What do you think about the person who made the comment? What do you think are the reasons why this person made that comment? | (self-generated) | – | n/a |

**Table 3 (continued) | Scales, item wording, and sources with grand means and standard deviations (Study 1 and Study 2)**

| Concept | Wording | Source | Study 1 M (SD) | Study 2 M (SD) |
|---|---|---|---|---|
| Resistance to change (S1 α = 0.90) | Indicate the extent to which you agree or disagree with each of the following statements. There are no right or wrong answers! 1 = strongly disagree, 7 = strongly agree Approaches used by people in the past are generally the most effective. If society is going to change, it should occur slowly and naturally. The established way of doing things should be protected and preserved. Fast or radical changes are unwise and dangerous. Traditions reflect wisdom and knowledge. Making sudden changes tends to create more problems than solutions. Slow, gradual change helps prevent catastrophes and mistakes. Established traditions are the best way to run society. | Adapted from White et al. (2020)[46] | 4.64 (1.29) | n/a |
| Reactance (S1 α = 0.91) | Indicate the extent to which you agree or disagree with each of the following statements. There are no right or wrong answers! 1 = disagree completely, 7 = agree completely Regulations trigger a sense of resistance in me. When something is prohibited, I usually think, "That's exactly what I am going to do." I consider advice from others to be an intrusion. I become frustrated when I am unable to make free and independent decisions. I become angry when my freedom of choice is restricted. Advice and recommendations usually induce me to do just the opposite. It makes me angry when another person is held up as a role model for me to follow. When someone forces me to do something, I feel like doing the opposite. | Adapted from Hong & Faedda (1996)[47] | 3.91 (1.50) | n/a |
| Engagement in pro-environmental behaviour (S1 α = 0.75; S2 α = 0.71) | Now, please respond to these questions about your behaviour. Don't feel any pressure, just indicate what you choose to do. 1 = Never, 7 = Always How often do you walk, bicycle, carpool, or take public transportation instead of driving a vehicle by yourself? How often do you turn your personal electronics off or in low-power mode when not in use? How often do you act to conserve water, when showering, cleaning clothes, dishes, watering plants, or other uses? When you are in PUBLIC, how often do you sort trash into the recycling? When you are in PRIVATE, how often do you sort trash into the recycling? How often do you carry a reusable water bottle? | Adapted from Brick et al. (2017)[48] | 4.85 (1.28) | 4.38 (1.30) |
| Conspiracy mentality [without rational suspicion] (S1 α = 0.93; S2 α = 0.93) | On this page, you will find beliefs about the state of affairs in the world. People agree or disagree to various extents with these beliefs. Indicate your agreement or disagreement with each statement on a 1 (=strongly disagree) to 7 (=strongly agree) scale. Do not overthink your answers; go with your initial hunches. Select strongly agree so that we are sure you are reading attentively [attention check] The government or covert organizations are responsible for events that are unusual or unexplained. The alternative explanations for important societal events are closer to the truth than the official story. Many so called "coincidences" are in fact clues as to how things really happened. Events throughout history are carefully planned and orchestrated by individuals for their own betterment. Many situations or events can be explained by illegal or harmful acts by the government or other powerful people. Some things that everyone accepts as true are in fact hoaxes created by people in power. Events on the news may not have actually happened. | Adapted from Stojanov & Halberstadt (2019)[49] | 4.29 (1.52) | 3.77 (1.48) |
| Trust in science (S1 α = 0.92; S2 α = 0.92) | On this survey, when we say 'science' we mean the understanding we have about the world from observation and testing. When we say 'scientists' we mean people who study the Planet Earth, nature, and medicine, among other things. Please answer truthfully to the questions below. 1 = Not at all, 7 = A lot How much do you trust scientists in this country? In general, would you say that you trust science? In general, how much do you trust scientists to find out accurate information about the world? How much do you trust scientists working in colleges/universities in this country to do each of the following? To do their work with the intention of benefiting the public. To be open and honest about who is paying for their work. Now, thinking about companies - for example, those who make medicines or agricultural supplies - how much do you trust scientists working for companies in this country to do each of the following? To do their work with the intention of benefiting the public. To be open and honest about who is paying for their work. | Adapted from Sturgis et al. (2021)[45] | 5.06 (1.30) | 5.10 (1.26) |

S1: Study 1; S2: Study 2. Scales with two αs are dependent measures that were completed twice, once after each target. The first value represents Round 1 and the second value represents Round 2.

**Study 2.** Environmental activists were evaluated as lower in expertise (H1a) and their findings as less credible (H1b) than their non-activist counterparts. Environmental scientists engaging in disruptive activism (vs. not engaging in activism at all) were also evaluated to be more hypocritical, confirming Hypothesis 2. Effects regarding Hypotheses 1 and 2 were medium in size and thus larger than in Study 1 (see Tables 5 and 6). Finally, in line with Hypothesis 3, engaging in disruptive activism (vs. not engaging in activism) led to lower trustworthiness evaluations of the described scientist's research field.

**Table 4 | Stimulus texts by scientist activist vs. scientist non-activist condition and message content counterbalancing (Study 2)**

| Condition and stimulus text | Version 1 | Version 2 |
|---|---|---|
| Activist condition, consumer products | "[Name of researcher] has been a researcher in environmental science for over a decade, focusing on how consumer products drive climate change. When she/he is not in the lab, Dr. [last name of researcher] devotes her/his time to activism, helping to organize street protests outside of the offices of consumer product companies. She/He was recently arrested along with a group of environmentalists trying to block the railroad cars bringing single-use plastic consumer goods into a shipping port. She/He is urging individuals to take drastic action to avoid the worst possible consequences of climate change." | "[Name of researcher] has been studying environmental sustainability for 15 years to understand the impact of consumer products on greenhouse gas emissions. When she/he is not in the lab, Dr. [last name of researcher] devotes her/his time to activism, helping to organize street protests outside of the offices of consumer product companies. She/He was recently arrested along with a group of environmentalists trying to block the railroad cars bringing single-use plastic consumer goods into a shipping port. She/He is urging individuals to take drastic action to avoid the worst possible consequences of climate change." |
| Activist condition, energy consumption | "[Name of researcher] has been a researcher in environmental science for over a decade, focusing on how energy consumption drives climate change. When she/he is not in the lab, Dr. [last name of researcher] devotes her/his time to activism, helping to organize street protests outside of the offices of fossil fuel companies. She/He was recently arrested along with a group of environmentalists trying to block the railroad cars bringing fracked methane gas into a shipping port. She/He is urging individuals to take drastic action to avoid the worst possible consequences of climate change." | "[Name of researcher] has been studying environmental sustainability for 15 years to understand the impact of energy consumption on greenhouse gas emissions. When she/he is not in the lab, Dr. [last name of researcher] devotes her/his time to activism, helping to organize street protests outside of the offices of fossil fuel companies. She/He was recently arrested along with a group of environmentalists trying to block the railroad cars bringing fracked methane gas into a shipping port. She/He is urging individuals to take drastic action to avoid the worst possible consequences of climate change." |
| Non-activist condition, consumer products | "[Name of researcher] has been a researcher in environmental science for over a decade, focusing on how consumer products drive climate change. Beyond her/his own research activities, Dr. [last name of researcher] is busy teaching classes on environmental science to undergraduate and graduate students. She/He recently organized an environmental science lecture series to bring together a group of researchers from different fields to better understand climate change. Whenever students or colleagues approach her/him, Dr. [last name of researcher] openly answers questions on the causes and consequences of climate change and provides evidence-based information." | "[Name of researcher] has been studying environmental sustainability for 15 years to understand the impact of consumer products on greenhouse gas emissions. Beyond her/his own research activities, Dr. [last name of researcher] is busy teaching classes on environmental science to undergraduate and graduate students. She/He recently organized an environmental science lecture series to bring together a group of researchers from different fields to better understand climate change. Whenever students or colleagues approach her/him, Dr. [last name of researcher] openly answers questions on the causes and consequences of climate change and provides evidence-based information." |
| Non-activist condition, energy consumption | "[Name of researcher] has been a researcher in environmental science for over a decade, focusing on how energy consumption drives climate change. Beyond her/his own research activities, Dr. [last name of researcher] is busy teaching classes on environmental science to undergraduate and graduate students. She/He recently organized an environmental science lecture series to bring together a group of researchers from different fields to better understand climate change. Whenever students or colleagues approach her/him, Dr. [last name of researcher] openly answers questions on the causes and consequences of climate change and provides evidence-based information." | "[Name of researcher] has been studying environmental sustainability for 15 years to understand the impact of energy consumption on greenhouse gas emissions. Beyond her/his own research activities, Dr. [last name of researcher] is busy teaching classes on environmental science to undergraduate and graduate students. She/He recently organized an environmental science lecture series to bring together a group of researchers from different fields to better understand climate change. Whenever students or colleagues approach her/him, Dr. [last name of researcher] openly answers questions on the causes and consequences of climate change and provides evidence-based information." |

Each participant read two texts, one in each type and one in each condition. The assignment of the type, the order, and the content was fully randomized.

As pre-registered, we further explored the effect of scientist activism on several additional dependent measures (Table 6). Engaging in activism reduced perceptions of scientists' integrity-based trust, benevolence-based trust, and their apparent environmental concern, although the latter effect was small. Finally, scientists who engaged (vs. did not engage) in activism led participants to report a lower intention to act pro-environmentally.

**Moderation analyses**

To explore potential moderators of the observed effects of scientists' activism on their perceived competence and hypocrisy, we included condition, the respective moderator, and the respective dependent variable (competence and hypocrisy) in a generalised linear model, including the participant as a random intercept. We followed up on significant Condition × Moderator interactions with Johnson-Neyman analyses to identify regions of significance (Figs. 2 & 3). Johnson-Neyman analyses consider the complete range of moderator values and indicate at which of these values a significant condition effect was observed. Accordingly, Johnson-Neyman analyses yield more unbiased information than do classic approaches (e.g., simple slopes) that focus only on a few arbitrary values of the moderator[37]. Here, we only report significant moderations (see Supplementary for a full report of the conducted analyses).

**Study 1**. Condition effects on competence evaluations were consistently moderated by reported environmental behaviour, $B = 0.15$, $SE = 0.04$,

$t(489) = 3.69$, $p < 0.001$, trust in science, $B = 0.15$, $SE = 0.04$, $t(489) = 3.71$, $p < 0.001$, and conspiracy mentality, $B = 0.10$, $SE = 0.04$, $t(489) = 2.71$, $p = 0.007$. Johnson-Neyman analyses indicated that the condition effect on competence was not significant for participants reporting high levels of pro-environmental behaviour (above 5.51), a very high trust in science (above 5.73), or a high conspiracy mentality (above 5.27, Fig. 2). We did not observe parallel moderation effects on hypocrisy.

Regarding demographics, condition effects on hypocrisy (but not on competence) were moderated by age and condition effects on competence (but not on hypocrisy) were moderated by self-description as an activist, such that younger participants (<39 years) and those who identified as activists exhibited no significant condition effect (Fig. 2).

**Study 2**. As pre-registered, we explored if the expected effects (i.e., H1-H3) were moderated by participants' own pro-environmental behaviour, conspiracy mentality, trust in science, or demographics (political orientation, age, gender, participants' identification as activists and scientists). Condition effects on all pre-registered dependent measures were moderated by reported environmental behaviour: expertise-based trust, $B = 0.12$, $SE = 0.04$, $t(634) = 3.03$, $p = .003$, hypocrisy, $B = -0.18$, $SE = 0.04$, $t(634) = -4.43$, $p < 0.001$, credibility, $B = 0.27$, $SE = 0.07$, $t(634) = 3.72$, $p < 0.001$, and trust in researcher's field, $B = 0.15$, $SE = 0.04$, $t(634) = 4.15$, $p < 0.001$. Trust in science also emerged as a consistent moderator of the condition effects on expertise-based trust, $B = 0.17$,

**Table 5 | Means, standard deviations, and test statistics of message source and message evaluations across conditions (Study 1)**

| Measure | Scientist Activist | Scientist Non-activist | t(490) | p | Effect size (d) | 95%CI (d) |
|---|---|---|---|---|---|---|
| Hypocrisy | 2.95 (1.87) | 2.77 (1.81) | −3.41 | 0.001 | −0.10 | [−0.16; −0.04] |
| Competence | 5.44 (1.53) | 5.65 (1.36) | 4.07 | < 0.001 | 0.15 | [0.08; 0.22] |
| Morality | 5.53 (1.51) | 5.55 (1.41) | 0.37 | 0.711 | 0.01 | [−0.06; 0.09] |
| Friendliness | 4.89 (1.62) | 4.92 (1.53) | 0.50 | 0.614 | 0.02 | [−0.05; 0.09] |
| Assertiveness | 5.75 (1.40) | 5.70 (1.35) | −0.86 | 0.392 | −0.03 | [−0.10; 0.04] |
| Constructiveness | 5.13 (1.63) | 5.28 (1.50) | 2.36 | 0.019 | 0.10 | [0.02; 0.18] |
| Attribution of environmental concern | 5.67 (1.42) | 5.70 (1.31) | 0.74 | 0.459 | 0.03 | [−0.05; 0.10] |
| Issue salience | 5.32 (1.52) | 5.37 (1.47) | 0.92 | 0.356 | 0.03 | [−0.03; 0.09] |
| Agreement with the comment | 5.35 (1.54) | 5.43 (1.47) | 1.61 | 0.107 | 0.05 | [−0.01; 0.11] |
| Sample size (n) | 491 | 491 | | | | |

Standard deviations are in parentheses. Uncorrected p-values are displayed; corrected α = 0.006.

**Table 6 | Means, standard deviations, and test statistics of person and research field evaluations across conditions (Study 2)**

| Measure | Scientist Activist | Scientist Non-activist | t(635) | p | Effect size (d) | 95%CI (d) |
|---|---|---|---|---|---|---|
| Hypocrisy (H2) | 2.49 (1.59) | 1.75 (1.11) | −13.54 | < 0.001 | −0.52 | [−0.60; −0.44] |
| Expertise-based trust (H1a) | 5.42 (1.45) | 6.24 (1.05) | 15.47 | <0.001 | 0.63 | [0.54; 0.72] |
| Integrity-based trust | 5.33 (1.56) | 5.98 (1.08) | 11.44 | <0.001 | 0.47 | [0.39; 0.56] |
| Benevolence-based trust | 5.07 (1.72) | 6.07 (1.10) | 15.44 | <0.001 | 0.67 | [0.57; 0.76] |
| Credibility (H1b) | 6.71 (2.94) | 8.15 (2.06) | 15.17 | <0.001 | 0.54 | [0.47; 0.62] |
| Action intent | 6.31 (3.06) | 6.99 (2.72) | 7.67 | <0.001 | 0.23 | [0.17; 0.29] |
| Attribution of environmental concern | 5.96 (1.37) | 6.19 (1.10) | 4.91 | <0.001 | 0.19 | [0.11; 0.26] |
| Trust in the researchers' field (H3) | 5.13 (1.66) | 5.52 (1.39) | 8.02 | <0.001 | 0.25 | [0.19; 0.31] |
| Sample size (n) | 636 | 636 | | | | |

Standard deviations are in parentheses. Uncorrected p-values are displayed; corrected α = .010.

$SE = 0.04$, $t(634) = 4.07$, $p < 0.001$, hypocrisy, $B = -0.20$, $SE = 0.04$, $t(634) = -4.78$, $p < 0.001$, credibility, $B = 0.34$, $SE = 0.07$, $t(634) = 4.55$, $p < 0.001$, and trust in researcher's field, $B = 0.16$, $SE = 0.04$, $t(634) = 4.11$, $p < 0.001$. Participants engaging in more pro-environmental behaviour and with higher trust in science showed smaller negative effects of scientists' activism. Johnson-Neyman analyses indicated that the condition effect on trust in the researchers' field was not significant for participants reporting very high levels of pro-environmental behaviour (above 5.96) or a very high trust in science (above 6.64, Fig. 3). No other transition points were observed.

Regarding demographics, condition effects on all pre-registered dependent measures were moderated by participants' political orientation. Johnson-Neyman analyses indicated that more left-leaning participants showed smaller negative effects of scientists' activism. Significant transition points emerged on hypocrisy (<1.18), credibility (<0.43) and the trust in the researchers' field (<1.82; Fig. 3). Condition effects on credibility (but not the other pre-registered dependent measures, expertise-based trust, hypocrisy, trust in researcher's field) were moderated by participants' age, $B = -0.02$, $SE = 0.01$, $t(631) = -2.86$, $p = .004$. Older participants showed a larger condition effect on credibility, but no transition points were observed (Fig. 3).

Finally, condition effects on all pre-registered dependent measures were moderated by participants' self-description as an activist (yes vs. no). A stronger Source Activism effect emerged among non-activist participants than among activist participants on expertise-based trust, $F(1, 617) = 13.62$, $p < 0.001$, $\eta^2_p = 0.02$, 90% CI [0.01, 0.04], hypocrisy, $F(1, 617) = 20.10$, $p < 0.001$, $\eta^2_p = 0.03$, 90% CI [0.01, 0.06], credibility, $F(1, 617) = 26.08$, $p < 0.001$, $\eta^2_p = 0.04$, 90% CI [0.02, 0.07], and trust in researcher's field, $F(1, 617) = 20.26$, $p < 0.001$, $\eta^2_p = 0.03$, 90% CI [0.01, 0.06] (Table 7).

## Discussion

There is an ongoing debate about the reputational costs and persuasive benefits of engaging in activism for environmental scientists, but experimental research on this topic is limited. To help fill this empirical gap, Study 1 compared community perceptions of environmental scientists who engaged in non-disruptive activism with those who engaged in public science communication. In Study 2, we compared perceptions of environmental scientists who engaged in disruptive activism with a non-activist control.

Overall, scientists were evaluated quite positively, as indicated by high average ratings of competence, morality, and trust as well as low hypocrisy (Table 3). Nevertheless, both studies detected negative effects of activism on perceptions of the target scientists' competence and hypocrisy. Study 2 revealed that these negative effects extended to participants' reduced trust in the field of the researcher and exploratory evidence of reduced intent to act on the researchers' findings, as well as other evaluations of trust and environmental concern. The observed effects in Study 2 were not only more consistent but also substantially larger than in Study 1, potentially due to the more extreme forms of activism described in the manipulation. In sum, engaging in activism had reliable negative effects on the perception of scientists and (in the case of disruptive activism) some effects on the effectiveness of their message and perception of their field. In both studies, these effects were particularly pronounced among those members of the community who need to be reached to have an impact: those with relatively low trust in science and low levels of pro-environmental behaviour.

Our findings stand in contrast to past experiments that did not observe negative consequences of scientists' activism on the credibility of their work or participants' intent to engage in climate action[24,25]. However, neither of these past experiments included no-activism control conditions but rather

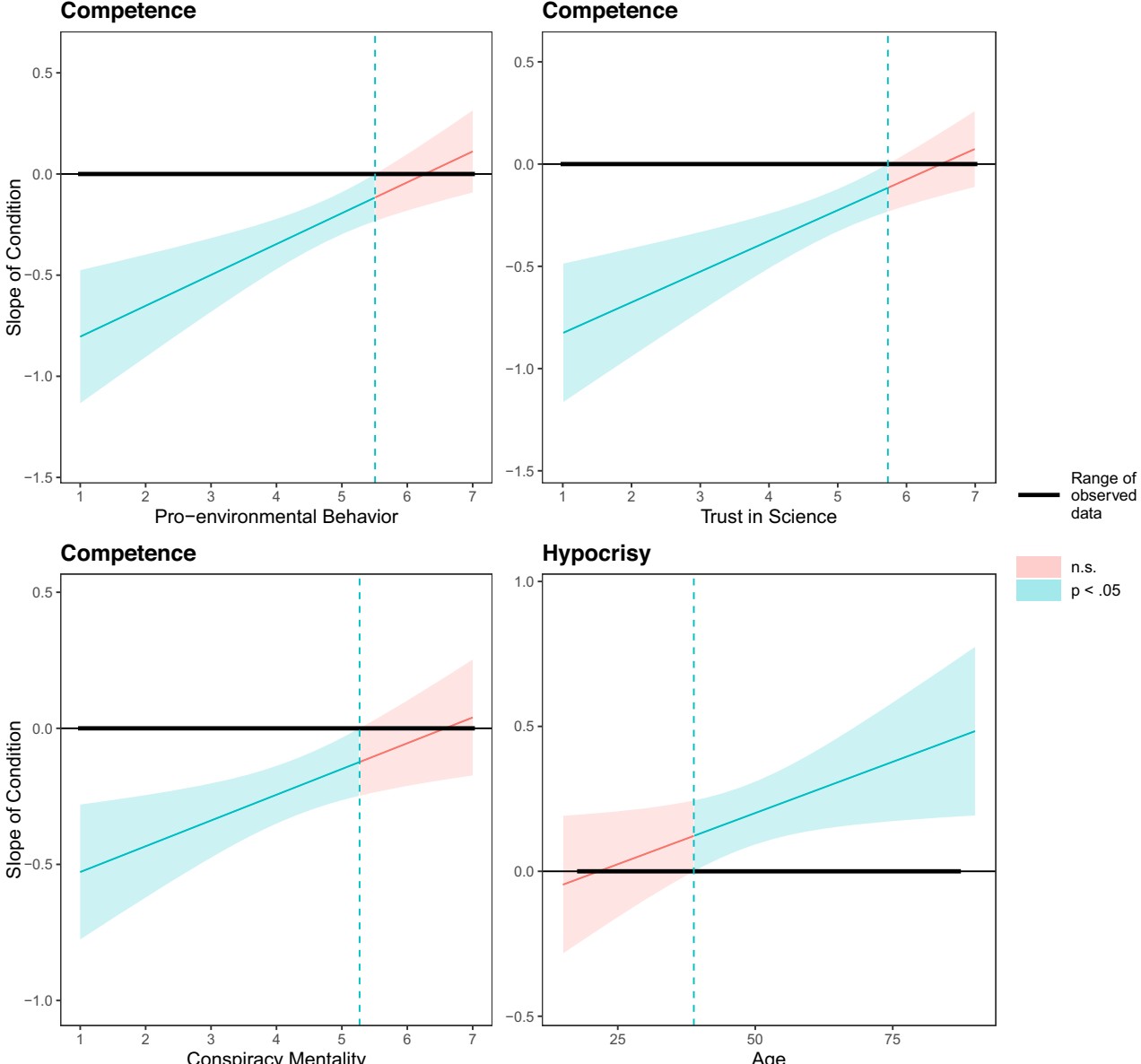

**Fig. 2 | Johnson-Neyman analyses of significant moderation effects (Study 1).** The message source effect (scientist activist vs. scientist non-activist) was not significant in red-shaded areas and significant at *p* <0.05 in the green-shaded areas (*N*=491 participants for pro-environmental behaviour, trust in science, and conspiracy mentality; *N* = 486 participants for age [5 participants did not indicate their age]). The y-axis indicates the observed effect size of the activist vs. non-activist scientist manipulation. Values above 0 indicate that evaluation values on the dependent variable were greater for activists than non-activists; values below 0 indicate that values on the dependent variable were lower for activists than non-activists. Thick horizontal line indicates the range of observed values of the moderator; dashed green lines indicate transition points (i.e., moderator values where the condition effect transitions between significant and non-significant).

contrasted different forms of advocacy. For instance, Friedman[25] observed no negative effects of disruptive activism over other types of civic engagement (i.e., non-disruptive activism or advocacy) on scientists' credibility and participants' intent to act. In contrast, we sought to support scientists who wish to make an informed decision whether to get actively involved in activism or not and thus included a "pure" no-activism control condition.

In fact, our Study 2 used Friedman's disruptive activism manipulation and some of their dependent measures. Interestingly, our participants responded to the disruptive activism description (credibility *M* = 6.71, *SD* = 2.94, action intent *M* = 6.31, *SD* = 3.06) in a manner similar to Friedman's participants (credibility *M* = 6.11, *SD* = 1.66, action intent *M* = 5.88, *SD* = 2.23). However, our control condition without activism, which was not included in Friedman's study, yielded significantly more positive ratings (credibility *M* = 8.15, *SD* = 2.06, action intent *M* = 6.99,

*SD* = 2.72). Apparently, activism reduced positive perceptions of scientists rather than triggering their rejection.

**Limitations and Caveats**
A few caveats should be noted. First, across both experimental conditions, evaluations of the scientists were generally positive, indicating that we are observing shifts within an overall favourable impression rather than outright rejection or hostility toward activist scientists. The effects, therefore, reflect relative differences in warmth and credibility rather than categorical disapproval. Second, public approval is not the sole metric of effective activism. Those engaging in activism may, in fact, not aim for approval but seek disruption to signal power[38], build momentum[39], or attract media attention[40]. The assumption is that the resulting tension is required for societal change—even if it comes at the cost of alienating some members of

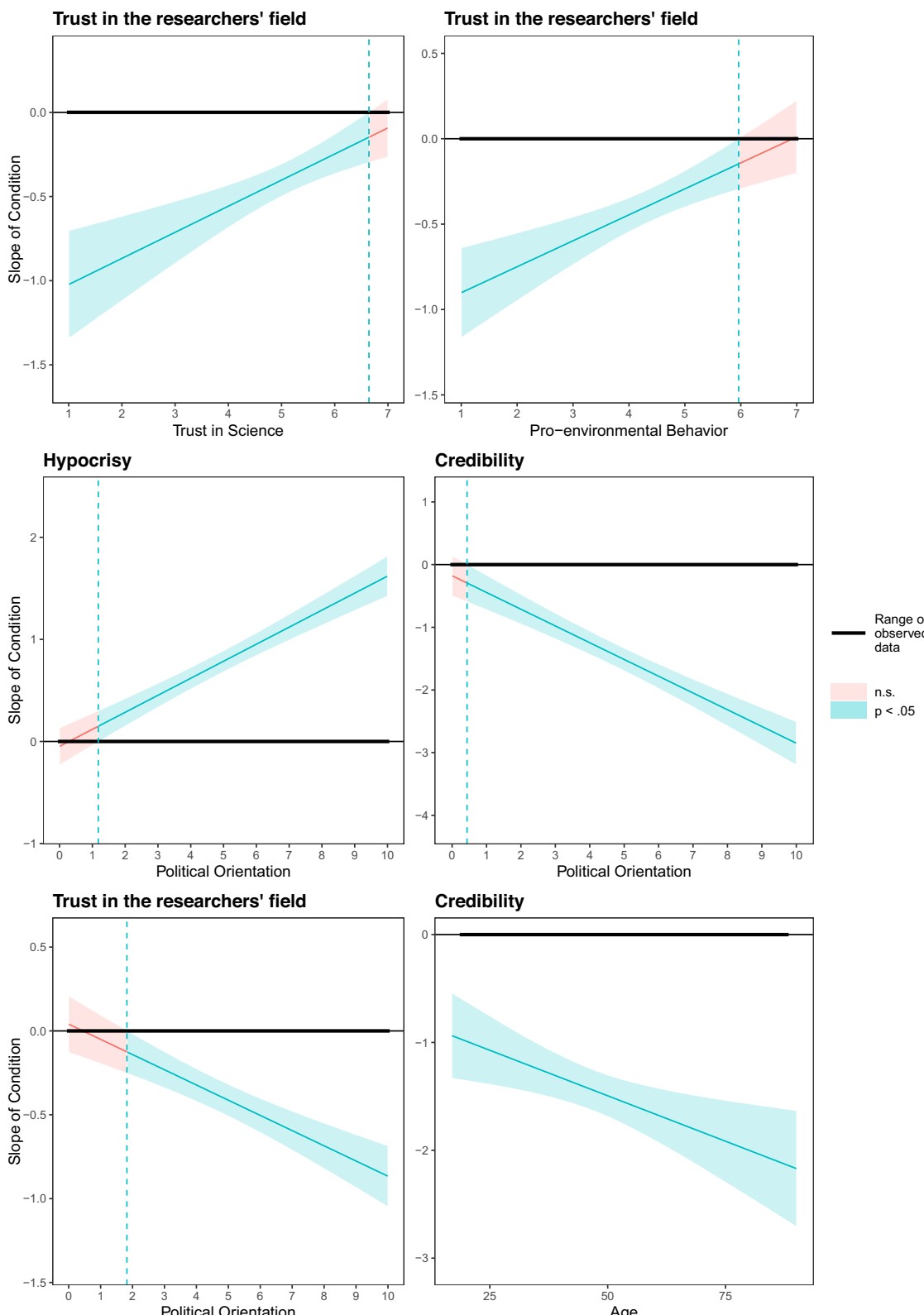

**Fig. 3 | Selected Johnson-Neyman analyses of significant moderation effects (Study 2).** The source activism effect (scientist activist vs. scientist non-activist) was not significant in red-shaded areas and significant at $p < 0.05$ in the green-shaded areas ($N = 636$ participants for pro-environmental behaviour, trust in science, and political orientation, $N = 633$ participants for age [3 participants did not indicate their age]). The y-axis indicates the observed effect size of the activist vs. non-activist scientist manipulation. Values above 0 indicate that evaluation values on the dependent variable were greater for activists than non-activists; values below 0 indicate that values on the dependent variable were lower for activists than non-activists. A thick horizontal line indicates the range of observed values of the moderator; dashed green lines indicate transition points (i.e., moderator values where the condition effect transitions between significant and non-significant). See Supplementary for all Johnson-Neyman graphs.

**Table 7 | Means, standard errors, and pairwise comparisons for hypocrisy, expertise-based trust, credibility, and trust in the researchers' field by participant activism and source conditions (Study 2)**

| Measure | Participant non-activist | | Participant activist | |
|---|---|---|---|---|
| | Scientist non-activist | Scientist activist | Scientist non-activist | Scientist activist |
| Hypocrisy | 1.86 (0.05)$_a$ | 2.79 (0.08)$_b$ | 1.53 (0.08)$_c$ | 1.94 (0.11)$_a$ |
| Expertise-based trust | 6.13 (0.05)$_a$ | 5.16 (0.07)$_b$ | 6.45 (0.07)$_c$ | 5.90 (0.10)$_a$ |
| Credibility | 7.79 (0.10)$_a$ | 5.98 (0.14)$_b$ | 8.88 (0.14)$_c$ | 8.10 (0.20)$_a$ |
| Trust in the researchers' field | 5.32 (0.07)$_a$ | 4.76 (0.08)$_b$ | 5.93 (0.10)$_c$ | 5.84 (0.11)$_c$ |

Standard errors are in parentheses. Cells sharing a subscript do not differ significantly at Bonferroni-adjusted α = 0.05.

the public. According to this view, disruptive and divisive actions may nonetheless be consequential if they attract media attention, shift public agendas, or pressure decision-makers to act (see, e.g., Ostarek, et al.[41]).

From this perspective, activism can have instrumental or environmental benefits even when it provokes ambivalence or resistance in the community, highlighting that reputational costs and social impact do not always move in tandem. Alternatively, activist events may serve an important function for the climate community, such as helping create a joint identity or attracting new members to join the movement. Whether increased media attention and a large, strongly-identified climate movement increase readiness for change in the broader population are important questions, but this was outside the auspices of the current study. We hope that the current research will help scientists to make an informed decision on whether to engage in activism.

Both reported studies were conducted in the US, a context highly polarized on science and environmental issues. For instance, Cologna et al.[22] observed that while average endorsement of scientists' public engagement was moderate in the US, these ratings were strongly correlated with participants' reported trust in science. This relationship between trust in science and the endorsement of scientists' engagement was weaker or even reversed in other countries. It is thus possible that the observed effects would be attenuated in contexts where science and the environment are less polarizing topics. Accordingly, future research should assess how engaging in activism influences the perception of scientists across cultures.

## Outlook

To resolve the activist dilemma described here—balancing the need to create change with the need to maintain a reputation as non-hypocritical and competent—scientists who wish to engage in activism have several options. One option is to separate science and activism activities by highlighting one's role as a private citizen, not an expert, during activism. This approach is generally in line with the pledge to only voice recommendations in line with one's own scientific expertise and in a non-defensive, neutral manner[42], but see[5]. Scientists could also choose to engage in advocacy, working behind the scenes for policy change with political actors they have unique access to, although advocacy may come with its own challenges[17-20]. To overcome these challenges, related research on advocacy shows that taking the position of an honest broker of information rather than emphasizing the epistemic implications of their knowledge for political decisions (i.e., an epistocrat) can go a long way[43]. Similar approaches may help attain the key goal to enhance the benefits of scientists' engagement in activism (e.g., provide skills, access to public stakeholders) while securing their scientific reputation.

In sum, our results show that, all else being equal, engaging in activism had moderate but reliable costs in terms of reduced competence and increased hypocrisy, as well as some small costs on downstream trust and perceptions of science. We hope our research will help scientists make

informed decisions on whether the benefits of activist engagement outweigh the potential costs.

## Data availability
Anonymized raw data are available at https://osf.io/pshak/.

## Code availability
Code for analyses is available at https://osf.io/pshak.

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

## Acknowledgements

This research was funded in part by the Austrian Science Fund (FWF) [https://doi.org/10.55776/P37261] awarded to J.L.T and ARC Laureate fellowship awarded to M.J.H. (FL230100022). The funders had no role in study design, data collection and analysis, decision to publish or preparation of the manuscript. We thank Julia Prohaska for her assistance in preparing the manuscript.

## Author contributions

Conceptualization, J.L.T., S.M.M., & M.J.H.; methodology, J.B. & J.L.T.; formal analysis, J.B.; resources, M.J.H & J.L.T.; data curation, J.B.; writing—original draft preparation, J.L.T. & J. B.; writing—review and editing, M.J.H & S.M.M.; visualization, J.B.; supervision, J.L.T.; project administration, J.B.; funding acquisition, M.J.H & J.L.T. All authors have read and agreed to the published version of the manuscript.

## Competing interests

The authors declare that they have no competing interests.
