## [Transparent Peer Review file · Communications Psychology]

Navigating the Credibility Risks of Environmental Scientists' Activism

Corresponding Author: Dr J. Lukas Thürmer

Version 0:

Decision Letter:

Dear Dr Thürmer,

Thank you for your patience during the peer-review process. Your manuscript titled "Navigating the Credibility Risks of Environmental Scientists' Climate Activism" has now been seen by 2 reviewers, whose comments are appended below. You will see that they find your work of some potential interest. However, they have raised quite substantial concerns that must be addressed. In light of these comments, we cannot accept the manuscript for publication, but would be interested in considering a revised version that fully addresses these serious concerns.

We hope you will find the Reviewers' comments useful as you decide how to proceed. Should additional work allow you to address these criticisms, we would be happy to look at a substantially revised manuscript. If you choose to take up this option, please highlight all changes in the manuscript text file, and provide a detailed point-by-point reply to the reviewers.

Editorially, we consider it crucial that additional empirical evidence is provided in the revised manuscript to address the reviewers' methodological concerns, such as stimulus materials and the choice of moderators. To this end, please include a new pre-registered, appropriately powered, study that provides strong evidence that supports the main conclusions of the study. Please also avoid over-interpretation of the results in the discussion section.

Please ensure you follow our statistical guidelines when reporting statistics (<https://www.nature.com/commspsychol/submit/submission-guidelines#statistical-guidelines>). Please note in particular our requirements for the reporting and interpretation of null-results. Non-significant findings derived from null-hypotheses significance tests should be reported in full, but may not be interpreted. Where you interpret null results, this interpretation must be based on Bayes Factors or equivalence tests.

I am attaching a checklist that details critical reporting requirements for the revised manuscript. Please attend to each item and ensure your manuscript is fully compliant. We are requesting that your manuscript aligns with these requirements as this facilitates the evaluation of your manuscript, reducing delays in re-review and potential future acceptance. If your revised manuscript is not aligned with these requests on major issues, such as those concerning statistics, it may be returned to you for further revisions without re-review. Additional information can be found in our style and formatting guide [a href="https://www.nature.com/documents/commspsychol-style-formatting-guide-accept.pdf">Communications Psychology formatting guide](https://www.nature.com/documents/commspsychol-style-formatting-guide-accept.pdf).

If the revision process takes significantly longer than five months, we will be happy to reconsider your paper at a later date, provided it still presents a significant contribution to the literature at that stage.

Please use the following link to submit your
- revised manuscript,
- point-by-point response to the referees' comments,
- cover letter (as a separate document),
- the Editorial Policy Checklist (see below),
- the Reporting Summary (see below), and
- the completed Editorial Request Table (attached):

Link Redacted

Thank you for the opportunity to review your work.

Best regards,

Troby Lui, on behalf of John Jamir Benzon Aruta, PhD (Editorial Board Member)

Troby Lui, PhD
Associate Editor
Communications Psychology

REVIEWER EXPERTISE:

Reviewer #1: trust in climate science and scientists

Reviewer #2: environmental psychology

REVIEWER REPORTS:

Reviewer #1 (Remarks to the Author):

The authors conducted a survey-embedded experiment to investigate perceptions of scientists who engage in activism and the persuasiveness of their messages. The authors address an important and understudied question. Indeed, there is a lack of empirical evidence on the impacts of scientists' activism and I agree with the authors that "In weighing up the potential credibility costs of engaging in activism, climate scientists have a surprisingly thin evidence base to draw on." Nevertheless, I have some concerns that need to be addressed before this manuscript can be considered for publication.

Major comments

1. Only a few studies have investigated how climate activism affects scientists' credibility. One study that is not mentioned in the manuscript, and which reaches a different conclusion, is a recent study by Friedman (2024), who found that the perception that a scientist had participated in civil disobedience did not undermine their credibility.

Friedman, R. S. (2024). Civil Disobedience by Environmental Scientists: An Experimental Study of its Influence on the Impact and Credibility of Climate Change Research. *Environmental Communication*, 18(4), 451–464.
<https://doi.org/10.1080/17524032.2024.2302532>

2. In the introduction, I recommend clearly differentiating between studies that focus on advocacy and those that focus on activism. I disagree with the sentence on lines 53-54 that the findings on advocacy are indicative of reservations about scientists' activism. First, many studies (including those cited by the authors), found no effect of policy advocacy on trust/credibility. Second, I am not aware of any study that assesses the impacts of both advocacy and activism on trust/credibility, which would allow to test differential impacts. It is plausible that activism would have different impacts on credibility than advocacy, which is why more studies on scientists' activism are needed.

3. The authors state: "A recent survey corroborates this view, as 4 out of 10 Americans indicated they expected scientists to focus on establishing sound scientific facts rather than contributing to policy debates." Please note that the most recent data from the PEW Research Center shows that 51% believe that scientists should take an active role in public policy debates about scientific issues: <https://www.pewresearch.org/science/2024/11/14/public-trust-in-scientists-and-views-on-their-role-in-policy-making/>

Recent global data from the TISP Many Labs study further shows that a majority of people worldwide believe that scientists should be involved in society and policymaking:

Cologna, V., Mede, N. G., Berger, S., Besley, J., Brick, C., Joubert, M., Maibach, E. W., Mihelj, S., Oreskes, N., Schäfer, M. S., van der Linden, S., Abdul Aziz, N. I., Abdulsalam, S., Shamsi, N. A., Aczel, B., Adinugroho, I., Alabrese, E., Aldoh, A., Alfano, M., ... Zwaan, R. A. (2025). Trust in scientists and their role in society across 68 countries. *Nature Human Behaviour*,

1–18. <https://doi.org/10.1038/s41562-024-02090-5>

4. Several studies on the impacts of scientists' advocacy on credibility are missing, for example:

Beall, L., Myers, T. A., Kotcher, J. E., Vraga, E. K., & Maibach, E. W. (2017). Controversy matters: Impacts of topic and solution controversy on the perceived credibility of a scientist who advocates. *PLOS ONE*, 12(11), e0187511. <https://doi.org/10.1371/journal.pone.0187511>

Post, S., & Bienzeisler, N. (2024). The Honest Broker versus the Epistocrat: Attenuating Distrust in Science by Disentangling Science from Politics. *Political Communication*, 0(0), 1–23. <https://doi.org/10.1080/10584609.2024.2317274>

5. How did the authors select the moderators to be included in this study? Please provide more information on the theoretical motivation behind the analytical choices.

6. Did the authors inspect whether self-describing as scientist or activist influenced the results?

7. How do the authors explain higher levels of perceived hypocrisy for the activist scientists? Some of the items such as “failing to practice the very same thing they preach” seem to be more in line with activist behavior. I was curious whether the open-ended questions could provide more insights on this. Analyzing the open-ended questions would further enrich the manuscript.

8. Relatedly, why did the authors assess competence and trust in scientists separately? In the trust literature, perceived competence or expertise are generally understood as a dimension of trustworthiness. Indeed, many scales measuring trust in scientists include measures of competence, see for example:

Hendriks, F., Kienhues, D., & Bromme, R. (2015). Measuring Laypeople's Trust in Experts in a Digital Age: The Muenster Epistemic Trustworthiness Inventory (METI). *PLOS ONE*, 10(10), e0139309. <https://doi.org/10.1371/journal.pone.0139309>

Cologna, V., Mede, N. G., Berger, S., Besley, J., Brick, C., Joubert, M., Maibach, E. W., Mihelj, S., Oreskes, N., Schäfer, M. S., van der Linden, S., Abdul Aziz, N. I., Abdulsalam, S., Shamsi, N. A., Aczel, B., Adinugroho, I., Alabrese, E., Aldoh, A., Alfano, M., ... Zwaan, R. A. (2025). Trust in scientists and their role in society across 68 countries. *Nature Human Behaviour*, 1–18. <https://doi.org/10.1038/s41562-024-02090-5>

Besley, J. C., Lee, N. M., & Pressgrove, G. (2020). Reassessing the Variables Used to Measure Public Perceptions of Scientists. *Science Communication*, 43(1), 3-32. <https://doi.org/10.1177/1075547020949547>

In general, there is a need for greater conceptual clarity. For example, in the abstract the word integrity is used when referring to competence and hypocrisy, but the term integrity does not appear in the main text. Integrity—just as competence—is generally understood as a dimension of trustworthiness that reflects honesty and is different from competence.

9. I recommend toning down the language in the discussion section, for example: “our highly controlled experiment”. I noticed that in one of the images people in the background wear masks, which could also influence perceptions. Further, many effect sizes (e.g., Cohen's $d = 0.18$) are negligible and this should be acknowledged.

10. The authors state: “It remains unclear what would happen if our stimuli incorporated more extreme forms of activism such as civil disobedience, although the social psychological literature in other contexts suggests that, if anything, this might exacerbate negative consequences for activists.” Please note that there is also a study that shows that disruptive protests can benefit the climate movement:

How disruptive climate protests can benefit the broader climate movement. *Nat Sustain* 7, 1564–1565 (2024). <https://doi.org/10.1038/s41893-024-01445-0>

Minor comments

1. On page 2, you state that: “A survey of authors and review editors of an IPCC report revealed that, although a significant proportion personally engaged in activism, nearly three-quarters believed the IPCC should refrain from climate advocacy.” This conflates the perceived role of individual scientists in activism with those of institutions such as the IPCC, which are by definition not policy-prescriptive.

2. It would be helpful if the authors provided a definition of “activism” in the beginning of the manuscript. Currently, examples of activism are only provided later in the text and terms such as “conventional forms of activism” are not very clear.

3. Why do the x-axes for pro-environmental behavior and trust in science go from 0-8 if these variables were assessed on a scale from 1-7? What do the dotted lines indicate?

Reviewer #2 (Remarks to the Author):

I appreciate the opportunity to review this manuscript, which poses a highly relevant and timely research question about the

causal influence of climate researchers' public-facing identities. The paper is well-written, and the introduction is clear and well-positioned within the broader literature. I also applaud the authors' open sharing of study materials.

Despite this, I am not overly unconvinced by the causal evidence provided for various reasons, including (1) the brevity and artificiality of the experimental materials and researcher descriptions, (2) the non-representative nature of the mTurk sample, and (3) the clearly AI-generated profile pictures. The paper also somewhat left the impression that the authors have already made up their minds about whether climate-related researchers should engage in activism (e.g., introduction and interpretation of results). Nevertheless, these concerns should not necessarily disqualify the paper from publication, but I believe some changes are pertinent before potential publication.

Researcher profiles

1. The logic behind the researcher profile descriptions requires more elaboration. Each profile includes many potentially influential elements, yet the rationale for choosing specific details is unclear. For example, why were the researcher's profiles focusing on energy use and waste reduction instead of, for example, climate science or climate policy? This may have influenced perceptions. All researcher profiles also appear to represent white individuals—was this a deliberate choice, and if so, what was the reasoning?
2. The profile pictures were visibly AI-generated. While I understand the practical constraints, the artificial nature of the images may have increased the perceived hypothetical nature of the study and influenced participants' evaluations. This should be articulated as a limitation in the discussion.
3. I find this phrasing in the activist framing problematic: "My research is closely aligned to my personal passions and my politics and I don't see a strong divide between my research world and my activist world." (p. 20). This wording implies that the researcher aligns their science with their ideology and politics. Yet, many scientists become activists because of concern for the societal implications of climate change based on scientific evidence. The current framing risks misrepresenting this motivation and could lead to biased responses against the 'activist' condition.
4. The term "neutral expert" is ambiguous in the context of climate change. For instance, a recent perspective article (van Eck et al., 2024) problematizes the notion of neutrality in climate expertise. The authors should clarify how they operationalize neutrality in their experimental condition and reflect on the normative implications of this framing. <https://www.nature.com/articles/s44168-024-00171-9>

Outcome and moderating variables

5. The theoretical and/or empirical rationale for selecting the many outcome variables and moderators is unclear. For example, why include an 8-item measure of perceived hypocrisy? Under what assumptions would one expect the activist researchers to be perceived as more hypocritical? I encourage the authors to clarify and justify the selection of at least some of the outcome variables and moderators.
6. While the materials and data are shared on OSF, the study appears not to have been preregistered. This is absolutely fine. However, if this is the case and combined with comment #5, the authors should consider labeling their analyses as exploratory and interpret findings appropriately.

Sample and analyses

7. Did you include a comprehension check to ensure participants read the target descriptions properly? If not, it may be interesting to control for time spent reading the profiles (e.g., if available in Qualtrics or whatever software was used).
8. The dropout rate (226 incomplete responses) is notable, raising concerns about selective attrition. Can the authors comment on the potential causes and whether attrition was systematically related to any variables of interest?
9. I highly value the power calculations and adjusting for multiple comparisons. However, the blanket statement that "power was sufficient to explore moderators" deserves elaboration, especially given the number of conducted moderation analyses.
10. It would be helpful to briefly summarize the descriptive results before discussing the experimental results. For example, the means for competence, morality, and agreement with the comment were relatively high across conditions.
11. Were there any differences between the energy-saving and waste-reduction researcher profiles? Given that both were used, this potential source of variation might be interesting to discuss.

Other comments

12. The abstract is too brief to adequately capture the study's focus and relevance. I encourage you to expand the abstract to provide more clarity on the research design, findings, and implications.
13. The authors use terminology inconsistently throughout the manuscript. I also had to read the method section several times to understand the experiment and measures properly. For example, you refer to the researcher's "comments" and

“messages” somewhat interchangeably, which confused me. Moreover, to me, these comments/messages essentially represent descriptions of researchers without making any clear comments. I worry that this also confused participants; for instance, questions like “What do you think about the person who made the comment?” and “To what extent do you think the person who wrote these comments cares about the environment?” might have been difficult to interpret. The latter obviously cannot be addressed but greater consistency and clarity in referring to the stimuli would be helpful.

14. Given the concerns outlined above, I believe the discussion of limitations is too superficial. I strongly encourage the authors to expand this section to reflect on limitations related to design, stimuli, sampling, and generalizability. For example, the sample is not representative of the US population, which should be highlighted (perhaps in addition to other quality concerns widely raised about mTurk).

EDITORIAL POLICIES

We ask that you ensure your manuscript complies with our editorial policies and reporting requirements.

To that end, we require revised manuscripts to be accompanied by two completed items: a reporting summary that collects information on study design and procedure, and an editorial policy checklist that verifies compliance with all required editorial policies

- <https://www.nature.com/documents/nr-reporting-summary.zip>>Nature Research Reporting Summary
- <https://www.nature.com/documents/nr-editorial-policy-checklist.pdf>>Editorial Policy Checklist

All points on the policy checklist must be addressed. Your revised manuscript can only be sent back to the referees if these checklists are completed and uploaded with the revision.

Notes: If you have submitted a Stage 1 Registered Report, Review, Primer, Comment, or Perspective you do not need to submit these forms. If you have already submitted these forms, you may disregard this request.

** Visit Nature Research's author and referees' website at <http://www.nature.com/authors>>www.nature.com/authors for information about policies, services and author benefits**

If you experience problems in linking your ORCID, please contact the <http://platformsupport.nature.com/>>Platform Support Helpdesk.

Version 1:

Decision Letter:

Dear Dr Thürmer,

Your manuscript titled "Navigating the Credibility Risks of Environmental Scientists' Activism" has now been seen by our reviewers, whose comments appear below. In light of their advice I am delighted to say that we are happy, in principle, to publish a suitably revised version in Communications Psychology.

We therefore invite you to revise your paper one last time to address the remaining concerns of our reviewers and a list of editorial requests. At the same time we ask that you edit your manuscript to comply with our format requirements and to maximise the accessibility and therefore the impact of your work.

EDITORIAL REQUESTS:

SUBMISSION INFORMATION:

OPEN ACCESS:

* DATA AVAILABILITY:

Link Redacted

Best regards,

Troy Lui, on behalf of John Aruta

Troy Lui, PhD
Associate Editor
Communications Psychology

John Jamir Benzon Aruta, PhD
Editorial Board Member
Communications Psychology
orcid.org/0000-0003-4155-1063

REVIEWERS' COMMENTS:

Reviewer #1 (Remarks to the Author):

I thank the authors for carefully addressing my points. I am impressed by the fact that the authors conducted an additional, pre-registered study to further substantiate and validate their initial findings. I agree with the other reviewer that in the initial submission it felt like the authors had already made up their minds about whether climate-related researchers should engage in activism. I found the revised version to be much more balanced and the limitations described more clearly and transparently. I have no further comments, thank the authors for their careful and substantial revision, and recommend publication.

Reviewer #2 (Remarks to the Author):

Thank you for carefully considering my comments. The manuscript has improved substantially, especially with the more extended introduction and addition of Study 2. I therefore only have a few minor comments:

1. I appreciate the authors' inclusion of descriptive results in Studies 1 and 2. However, I encourage adding a few sentences interpreting those results, such as commenting on the mean levels of competence, hypocrisy, morality, etc. This will help contextualize the experimental results. For example, is it a cause of concern that perceived hypocrisy slightly increases for activists versus non-activists if the mean perceived hypocrisy is still quite low?
2. Out of curiosity, what is the expected or potential causal mechanism for this finding in Study 2 – “Finally, scientists who engaged (vs. did not engage) in activism led participants to report a lower intention to act pro-environmentally.”?
3. This statement in the general discussion reflects a theory of change that, while logical, may not be universally true for achieving political change: “In both studies these effects were particularly pronounced among those members of the community who need to be reached to have an impact: those with relatively low trust in science and low levels of pro-environmental behaviour.” In the US, political change often happens without public support (e.g., anti-abortion legislation), and environmentally significant behavior change may occur through changes in economic incentives, infrastructure, etc., without necessarily changing people’s “minds” first.
4. In the general discussion, I felt there was a lack of reflection on the study context. To what extent do the results and their implications generalize to other contexts than the United States, given its unique political system and political climate?

Responses to Editor Comments

Associate Editor Comment 1

Thank you for your patience during the peer-review process. Your manuscript titled "Navigating the Credibility Risks of Environmental Scientists' Climate Activism" has now been seen by 2 reviewers, whose comments are appended below. You will see that they find your work of some potential interest. However, they have raised quite substantial concerns that must be addressed. In light of these comments, we cannot accept the manuscript for publication, but would be interested in considering a revised version that fully addresses these serious concerns.

We hope you will find the Reviewers' comments useful as you decide how to proceed. Should additional work allow you to address these criticisms, we would be happy to look at a substantially revised manuscript. If you choose to take up this option, please highlight all changes in the manuscript text file, and provide a detailed point-by-point reply to the reviewers.

Editorially, we consider it crucial that additional empirical evidence is provided in the revised manuscript to address the reviewers' methodological concerns, such as stimulus materials and the choice of moderators. To this end, please include a new pre-registered, appropriate powered, study that provides strong evidence that supports the main conclusions of the study. Please also avoid over-interpretation of the results in the discussion section.

Please ensure you follow our statistical guidelines when reporting statistics (<https://www.nature.com/commpsychol/submit/submission-guidelines#statistical-guidelines>). Please note in particular our requirements for the reporting and interpretation of null-results. Non-significant findings derived from null-hypotheses significance tests should be reported in full, but may not be interpreted. Where you interpret null results, this interpretation must be based on Bayes Factors or equivalence tests.

Author Response 1: Thank you for the opportunity to revise our manuscript. We have responded to all of your and the reviewers' comments, as detailed below.

We have conducted an additional, pre-registered study, in line with your and the reviewers' recommendations. Specifically, we employed the activism manipulation reported by Friedman (2024) and contrast it with a no-activism control condition. We also follow their established procedure and now use text rather than pictures as our manipulation. This pre-registered study fully replicated and extended the results of Study 1. Specifically, disruptive activism led to lower evaluations of expertise-based trust and competence, higher ratings of hypocrisy, and lower ratings of trust in the researcher's field. Beyond these registered analyses, exploratory test also showed that disruptive activism led to lower ratings of the other two dimensions of trust, benevolence and integrity, as well as lower attributions of environmental concern to the scientist and weaker intentions to act on the implications of their research.

In the revised general discussion, we now focus on observed findings and have carefully edited any statements that could be perceived to be over-interpretations. In doing so, we refrain from interpreting null results.

Response to Reviewer 1 Comments

Reviewer 1 Comment 1:

The authors conducted a survey-embedded experiment to investigate perceptions of scientists who engage in activism and the persuasiveness of their messages. The authors address an important and understudied question. Indeed, there is a lack of empirical evidence on the impacts of scientists' activism and I agree with the authors that "In weighing up the potential credibility costs of engaging in activism, climate scientists have a surprisingly thin evidence base to draw on." Nevertheless, I have some concerns that need to be addressed before this manuscript can be considered for publication.

Only a few studies have investigated how climate activism affects scientists' credibility. One study that is not mentioned in the manuscript, and which reaches a different conclusion, is a recent study by Friedman (2024), who found that the perception that a scientist had participated in civil disobedience did not undermine their credibility.

Friedman, R. S. (2024). Civil Disobedience by Environmental Scientists: An Experimental Study of its Influence on the Impact and Credibility of Climate Change Research. *Environmental Communication*, 18(4), 451–464. <https://doi.org/10.1080/17524032.2024.2302532>

Author Response 1: Thank you for serving as a reviewer on our manuscript and your constructive feedback. We originally did not discuss Friedman as it focusses on civil disobedience, a special and more disruptive type of activism. In the revised manuscript, we widen our focus, which now allows us to discuss these findings in detail. In fact, our new Study 2 builds on Friedman's methodology, including the activism manipulation as well as two of their dependent measures. We again observed reduced competence/expertise and increased hypocrisy ratings of activist vs. non-activist scientists. Interestingly, we also find consistent effects of engaging in disruptive activism (versus not engaging in activism) on two measures that yielded no effects in Friedman's study, reduced evaluations of scientists' credibility and participants' lower intent to act on the scientists' findings. We discuss how using a no-activism control condition rather than an advocacy control condition (in Friedman's study) may have contributed to these diverging findings.

Quote: "Our findings stand in contrast to past experiments that did not observe negative consequences of scientists' activism on the credibility of their work or participants' intent to engage in climate action.^{24, 25} However, neither of these past experiments included no-activism control conditions but rather contrasted different forms of advocacy. For instance, Friedman²⁵ observed no negative effects of disruptive activism over other types of civic engagement (i.e., non-disruptive activism or advocacy) on scientists' credibility and participants' intent to act. In contrast, we sought to support scientists who wish to make an informed decision whether to get actively involved in activism or not and thus included a "pure" no-activism control condition. In fact, our Study 2 used Friedman's disruptive activism manipulation and some of their dependent measures.

Interestingly, our participants responded to the disruptive activism description (credibility $M = 6.71$, $SD = 2.94$, action intent $M = 6.31$, $SD = 3.06$) similarly as Friedman's participants (credibility $M = 6.11$, $SD = 1.66$, action intent $M = 5.88$, $SD = 2.23$). However, our control condition without activism, which was not included in Friedman's study, yielded significantly more positive ratings (credibility $M = 8.15$, $SD = 2.06$, action intent $M = 6.99$, $SD = 2.72$). Apparently, activism reduced positive perceptions of scientists rather than triggering their rejection." (p. 21)

Reviewer 1 Comment 2:

In the introduction, I recommend clearly differentiating between studies that focus on advocacy and those that focus on activism. I disagree with the sentence on lines 53-54 that the findings on advocacy are indicative of reservations about scientists' activism. First, many studies (including those cited by

the authors), found no effect of policy advocacy on trust/credibility. Second, I am not aware of any study that assesses the impacts of both advocacy and activism on trust/credibility, which would allow to test differential impacts. It is plausible that activism would have different impacts on credibility than advocacy, which is why more studies on scientists' activism are needed.

Author Response 2: We now discuss the differences between advocacy and activism (see first Quote below) and structure our literature review accordingly (p. 3). We also touch on the different impact that advocacy and activism may have (see second Quote below), although a systematic test is beyond the scope of the current paper.

Quote: “In weighing up the potential credibility costs of engaging in activism, environmental scientists have a surprisingly thin evidence base to draw on. In evaluating the evidence, it is important to distinguish between advocacy and activism. Advocacy typically refers to activities in which scientists use their expertise to communicate, recommend, or support evidence-based solutions to policymakers or the public, usually within conventional institutional channels. Activism, in contrast, typically refers to direct public action—such as protests, demonstrations, or campaigns—that explicitly calls for social or political change and may be perceived as more confrontational. Although both are forms of public engagement, activism carries a stronger association with partisanship and moral urgency, which may make it more contested in terms of public perceptions of scientific credibility. (p. 3)

Quote: “Scientists could also choose to engage in advocacy, working behind the scenes for policy change with political actors they have unique access to, although advocacy may come with its own challenges.^{17, 18, 19, 20} To overcome these challenges, related research on advocacy shows that taking the position of an honest broker of information rather than emphasizing the epistemic implications of their knowledge for political decisions (i.e., an epistocrat) can go a long way.⁴¹ Similar approaches may help attain the key goal to enhance the benefits of scientists' engagement in activism (e.g., provide skills, access to public stakeholders) while securing their science reputation.” (p. 23)

Reviewer 1 Comment 3:

The authors state: “A recent survey corroborates this view, as 4 out of 10 Americans indicated they expected scientists to focus on establishing sound scientific facts rather than contributing to policy debates.” Please note that the most recent data from the PEW Research Center shows that 51% believe that scientists should take an active role in public policy debates about scientific issues: <https://www.pewresearch.org/science/2024/11/14/public-trust-in-scientists-and-views-on-their-role-in-policymaking/>

Recent global data from the TISP Many Labs study further shows that a majority of people worldwide believe that scientists should be involved in society and policymaking:

Cologna, V., Mede, N. G., Berger, S., Besley, J., Brick, C., Joubert, M., Maibach, E. W., Mihelj, S., Oreskes, N., Schäfer, M. S., van der Linden, S., Abdul Aziz, N. I., Abdulsalam, S., Shamsi, N. A., Aczel, B., Adinugroho, I., Alabrese, E., Aldoh, A., Alfano, M., ... Zwaan, R. A. (2025). Trust in scientists and their role in society across 68 countries. *Nature Human Behaviour*, 1–18. <https://doi.org/10.1038/s41562-024-02090-5>

Author Response 3: Thank you for pointing out this additional evidence, which we now discuss as follows:

Quote: “Others have observed that a slight majority of the overall U.S. population (51%) agrees that scientists should take an active role in public policy debates, although this agreement was substantially higher among Democrats (67%) than Republicans (35%).²¹ Globally, support for scientists' advocacy was observed at a comparable level (54%), with only 20% disagreeing or strongly disagreeing.²² Overall, these studies

suggest generally neutral-to-positive views about scientists' advocacy, although this support does not appear to be universal." (p. 4)

Reviewer 1 Comment 4:

Several studies on the impacts of scientists' advocacy on credibility are missing, for example:

Beall, L., Myers, T. A., Kotcher, J. E., Vraga, E. K., & Maibach, E. W. (2017). Controversy matters: Impacts of topic and solution controversy on the perceived credibility of a scientist who advocates. PLOS ONE, 12(11), e0187511. <https://doi.org/10.1371/journal.pone.0187511>

Post, S., & Bienzeisler, N. (2024). The Honest Broker versus the Epistocrat: Attenuating Distrust in Science by Disentangling Science from Politics. Political Communication, 0(0), 1–23. <https://doi.org/10.1080/10584609.2024.2317274>

Author Response 4: We now include Beall et al in the introduction (see Quote below). Post & Bienzeisler tested differences between different types of advocacy (epistocrat vs. honest broker) but did not include a no- advocacy condition. Accordingly, we address their findings in the general discussion (see second Quote in Response 2 above).

Quote: "A further experiment found that non-controversial advocacy (operationalised as writing Op-Eds presenting information and suggesting education campaigns) increased climate scientists' credibility compared to only providing scientific information, although this benefit disappeared when the advocacy included recommendations of regulation and mandating of solutions.¹⁸" (p. 4)

Reviewer 1 Comment 5:

How did the authors select the moderators to be included in this study? Please provide more information on the theoretical motivation behind the analytical choices.

Author Response 5: We thank the reviewer for this comment. Our inclusion of these moderators was motivated less by a single unifying theory and more by converging intuitions from prior research and the plausibility of these variables shaping how audiences respond to scientists' activism. We preregistered them as exploratory in Study 2 to reflect this more open-ended rationale and explicate our reasoning in the revised paper as follows:

Quote: "Both studies also included exploratory moderators of these effects, based on our intuition and past research. Specifically, we measured past environmental behaviour as people who already engage in pro-environmental behaviours may be more sympathetic to environmental scientists' activism than less engaged individuals. Similarly, we included measures of trust in science, as participants with higher trust in science may be more likely to extend this trust to scientists engaging in activism as compared to less trusting individuals. Conversely, we included conspiracy mentality as a higher conspiracy mentality could intensify scepticism toward scientists when they are seen as political actors. We also included trait-level reactance and resistance to change as exploratory moderators, reflecting the idea that responses to scientist activism may depend on people's general motivation to protect autonomy and stability. Those high in reactance may view activist scientists as overstepping or coercive, triggering defensive scepticism, whereas those high in resistance to change may oppose activism because it symbolises disruption to the social or political status quo." (p. 5)

Reviewer 1 Comment 6:

Did the authors inspect whether self-describing as scientist or activist influenced the results?

Author Response 6: We now include these analyses:

Quote in Study 1: “Regarding demographics, condition effects on hypocrisy (but not on competence) were moderated by age and self-description as an activist such that younger participants (< 39 years) and those who identified as activists exhibited no significant condition effect. We did not observe systematic interactions of participant gender, target sex, or participants’ identification as a scientist with condition (see Supplementary for details).” (p. 13)

Quote in Study 2: “Finally, condition effects on all pre-registered dependent measures were moderated by participants’ self-description as an activist (yes vs. no) such that a stronger Source Activism effect emerged among non-activist participants than among activist participants (Table 7). Participants’ self-description as a scientist (yes vs. no) did not emerge as a moderator.” (p. 20)

Reviewer 1 Comment 7:

How do the authors explain higher levels of perceived hypocrisy for the activist scientists? Some of the items such as “failing to practice the very same thing they preach” seem to be more in line with activist behavior. I was curious whether the open-ended questions could provide more insights on this. Analyzing the open-ended questions would further enrich the manuscript.

Author Response 7: We thank the reviewer for raising this point. We believe the finding that activist scientists were judged as more hypocritical can be understood in two ways. First, the very act of taking a strong public stance on an ethical issue increases exposure to hypocrisy judgments: by making explicit moral claims, the standard against which one’s personal behaviour is measured becomes more demanding, magnifying perceived inconsistencies. Second, prior work on “do-gooder derogation” (e.g., Monin’s work) suggests that individuals who adopt a morally elevated position can elicit defensive reactions, as audiences may feel implicitly criticised. Labelling activists as “hypocritical” may therefore function both as an assessment of potential inconsistency and as a means of reducing the moral threat posed by those who appear to claim the moral high ground. The free-text responses we obtained were rather short (11.69 words on average) and thus unfortunately precluded a meaningful content analysis

Reviewer 1 Comment 8:

Relatedly, why did the authors assess competence and trust in scientists separately? In the trust literature, perceived competence or expertise are generally understood as a dimension of trustworthiness. Indeed, many scales measuring trust in scientists include measures of competence, see for example:

Hendriks, F., Kienhues, D., & Bromme, R. (2015). Measuring Laypeople’s Trust in Experts in a Digital Age: The Muenster Epistemic Trustworthiness Inventory (METI). *PLOS ONE*, 10(10), e0139309. <https://doi.org/10.1371/journal.pone.0139309>

Cologna, V., Mede, N. G., Berger, S., Besley, J., Brick, C., Joubert, M., Maibach, E. W., Mihelj, S., Oreskes, N., Schäfer, M. S., van der Linden, S., Abdul Aziz, N. I., Abdulsalam, S., Shamsi, N. A., Aczel, B., Adinugroho, I., Alabrese, E., Aldoh, A., Alfano, M., ... Zwaan, R. A. (2025). Trust in scientists and their role in society across 68 countries. *Nature Human Behaviour*, 1–18. <https://doi.org/10.1038/s41562-024-02090-5>

Besley, J. C., Lee, N. M., & Pressgrove, G. (2020). Reassessing the Variables Used to Measure Public Perceptions of Scientists. *Science Communication*, 43(1), 3-32. <https://doi.org/10.1177/1075547020949547>

In general, there is a need for greater conceptual clarity. For example, in the abstract the word integrity is used when referring to competence and hypocrisy, but the term integrity does not appear in the main text. Integrity—just as competence—is generally understood as a dimension of trustworthiness that reflects honesty and is different from competence.

Author Response 8: We have revised the manuscript to enhance conceptual clarity and discuss how we assessed the perceptions of the scientists. To provide further empirical evidence regarding the evaluation of scientists, we assessed all three facets of trust using the METI in the new Study 2. We predicted and observed effects of activism on reduced expertise, parallel to the effects on competence in Study 1, and replicated the observed effects on hypocrisy. Interestingly, effects extended to the other trust dimensions as we observed reduced benevolence and integrity evaluations in the scientist activism condition in our exploratory analyses (see Manipulation Effects on p. 17 and Table 6 on p. 44).

Reviewer 1 Comment 9:

I recommend toning down the language in the discussion section, for example: “our highly controlled experiment”. I noticed that in one of the images people in the background wear masks, which could also influence perceptions. Further, many effect sizes (e.g., Cohen’s $d = 0.18$) are negligible and this should be acknowledged.

Author Response 9: We have revised the discussion sections and now acknowledge the small effects observed in Study 1 as follows:

Quote: “Study 1 suffers from three limitations. First, the study was not pre-registered and no formal predictions were made for these specific effects of competence and hypocrisy to emerge. Accordingly, these findings must be considered exploratory. Second, the activism described in the experimental condition was conventional (e.g., attending rallies). However, advocates for scientists’ activism commonly call for more disruptive forms of activism^{5, 6, 25} such as civil disobedience, which is the focus of Study 2. Third, our observed effects were small, potentially due to the materials or the control condition used: The AI-generated background-scenes may have been unconvincing for some participants. Likewise, the description of public engagement with the press in the non-activist scientist condition may have been interpreted as a form of activism by some participants.” (p. 13)

Reviewer 1 Comment 10:

The authors state: “It remains unclear what would happen if our stimuli incorporated more extreme forms of activism such as civil disobedience, although the social psychological literature in other contexts suggests that, if anything, this might exacerbate negative consequences for activists.” Please note that there is also a study that shows that disruptive protests can benefit the climate movement:

How disruptive climate protests can benefit the broader climate movement. *Nat Sustain* 7, 1564–1565 (2024). <https://doi.org/10.1038/s41893-024-01445-0>

Author Response 10: We now discuss the possible different consequences of engaging in more extreme forms of activism and provide an empirical test by including descriptions of disruptive activism in Study 2 (see Quote below). Interestingly, we again observed negative consequences of activism (as compared to no activism) and the effects were substantially larger in Study 2 than in Study 1.

Quote: “Second, Study 2 adapted the civil disobedience manipulation and measures from past work.²⁵ The consequences of such disruptive activism are somewhat unclear. Social psychological literature in other contexts suggests that more disruptive activism might exacerbate negative consequences for the perception of scientists as impartial brokers of information.³⁶ However, disobedience may also signal a greater urgency of the issue while preserving credibility evaluations, at least in a supportive audience (i.e., among college students).²⁵ Finally, disruptive activists may promote engagement with the cause even if they are personally evaluated negatively.³⁷” (p. 14)

Reviewer 1 Comment 11:

Minor comments

1. On page 2, you state that: “A survey of authors and review editors of an IPCC report revealed that, although a significant proportion personally engaged in activism, nearly three-quarters believed the IPCC should refrain from climate advocacy.” This conflates the perceived role of individual scientists in activism with those of institutions such as the IPCC, which are by definition not policy-prescriptive.
2. It would be helpful if the authors provided a definition of “activism” in the beginning of the manuscript. Currently, examples of activism are only provided later in the text and terms such as “conventional forms of activism” are not very clear.
3. Why do the x-axes for pro-environmental behavior and trust in science go from 0-8 if these variables were assessed on a scale from 1-7? What do the dotted lines indicate?

Author Response 11: 1. We have removed the reference to the IPCC.

2. We now provide the following definition of advocacy and activism:

Quote: “Advocacy typically refers to activities in which scientists use their expertise to communicate, recommend, or support evidence-based solutions to policymakers or the public, usually within conventional institutional channels. Activism, in contrast, typically refers to direct public action—such as protests, demonstrations, or campaigns—that explicitly calls for social or political change and may be perceived as more confrontational. Although both are forms of public engagement, activism carries a stronger association with partisanship and moral urgency, which may make it more contested in terms of public perceptions of scientific credibility.” (p. 3)

3. We have corrected the x-axes in all Johnson-Neyman plots and provide the following explanation for the dashed lines

Quote: “dashed green lines indicate transition points (i.e., moderator values where the condition effect transitions between significant and non-significant).” (p. 12 & p. 20)

Thank you again for serving as a reviewer on our manuscript and your constructive comments.

Response to Reviewer 2 Comments

Reviewer 2:

I appreciate the opportunity to review this manuscript, which poses a highly relevant and timely research question about the causal influence of climate researchers' public-facing identities. The paper is well-written, and the introduction is clear and well-positioned within the broader literature. I also applaud the authors' open sharing of study materials.

Despite this, I am not overly unconvinced by the causal evidence provided for various reasons, including (1) the brevity and artificiality of the experimental materials and researcher descriptions, (2) the non-representative nature of the mTurk sample, and (3) the clearly AI-generated profile pictures. The paper also somewhat left the impression that the authors have already made up their minds about whether climate-related researchers should engage in activism (e.g., introduction and interpretation of results). Nevertheless, these concerns should not necessarily disqualify the paper from publication, but I believe some changes are pertinent before potential publication.

Author Response: Thank you for serving as a reviewer and your constructive feedback. In response, we now discuss the limitations of our initial study (see Quote below) and have conducted a second study using an established text-based activism manipulation rather than the generated pictures. We also pre-registered the second study to increase the reliability of the findings and recruited a stratified Prolific Academic sample that was representative of the US population in terms of age, gender and political affiliation to increase the generalizability of our findings. In this research project, our aim is to provide a balanced discussion on the possible costs and benefits of engaging in activism (which some of us do) and revised the introduction accordingly.

Quote: "Study 1 suffers from three limitations. First, the study was not pre-registered and no formal predictions were made for these specific effects of competence and hypocrisy to emerge. Accordingly, these findings must be considered exploratory. Second, the activism described in the experimental condition was conventional (e.g., attending rallies). However, advocates for scientists' activism commonly call for more disruptive forms of activism^{5, 6, 25} such as civil disobedience, which is the focus of Study 2. Third, our observed effects were small, potentially due to the materials or the control condition used: The AI-generated background-scenes may have been unconvincing for some participants. Likewise, the description of public engagement with the press in the non-activist scientist condition may have been interpreted as a form of activism by some participants." (p. 13)

Reviewer 2 Comment 1:

Researcher profiles

1. The logic behind the researcher profile descriptions requires more elaboration. Each profile includes many potentially influential elements, yet the rationale for choosing specific details is unclear. For example, why were the researcher's profiles focusing on energy use and waste reduction instead of, for example, climate science or climate policy? This may have influenced perceptions. All researcher profiles also appear to represent white individuals—was this a deliberate choice, and if so, what was the reasoning?

Author Response 1: In revising the manuscript, we have broadened our focus to encompass all environmental activism. As such, we sought to include two research areas that both are relevant to protecting the environment and have equally strong practical implications. Moreover, these research areas afford engaging in expertise-based activism and/or advocacy but do not necessitate it (as one may assume climate policy research does). The selected stimulus pictures and texts include target sex as covariate (between participants). We considered including other demographics, such

race and age, but refrained to do so to reduce the variance in the stimulus materials. We highlight this methodological choice in the Study 1 method section as indicated below. In Study 2, we did not use visually identifiable pictures but rather relied on an established text manipulation. Results were remarkably consistent with Study 1. Apparently, the display modality in Study 1 did not drive the observed effects.

Quote: “Target person images were selected from the Chicago Face Database (CFD), a set of standardised and rated portrait pictures. Pictures were selected to reduce the variance in the stimulus materials, that is, all were Caucasian men and women, matched for average attractiveness, trustworthiness, and prototypicality, with a rated age range of 32.88 to 41.69 years. Four sets of faces were chosen (Fig. 1).” (p. 7)

Reviewer 2 Comment 2:

The profile pictures were visibly AI-generated. While I understand the practical constraints, the artificial nature of the images may have increased the perceived hypothetical nature of the study and influenced participants' evaluations. This should be articulated as a limitation in the discussion.

Author Response 2: We selected the profile pictures from real photos included in the Chicago Face Data Base, a collection of highly controlled portrait pictures of actual people. We selected the photos according to average attractiveness, trustworthiness, and prototypicality and an age around 35 years (see Quote above). The background scenes were AI generated, as we now discuss as a potential limitation (see Quote below).

Quote: “Third, our observed effects were small, potentially due to the materials or the control condition used: The AI-generated background-scenes may have been unconvincing for some participants. Likewise, the description of public engagement with the press in the non-activist scientist condition may have been interpreted as a form of activism by some participants.” (p. 13)

Reviewer 2 Comment 3:

I find this phrasing in the activist framing problematic: “My research is closely aligned to my personal passions and my politics and I don’t see a strong divide between my research world and my activist world.” (p. 20). This wording implies that the researcher aligns their science with their ideology and politics. Yet, many scientists become activists because of concern for the societal implications of climate change based on scientific evidence. The current framing risks misrepresenting this motivation and could lead to biased responses against the ‘activist’ condition.

Author Response 3: Our new Study 2 adapted the activism manipulation from previous research (Friedman, 2024) to circumvent these interpretational issues. We fully replicated the effects we observed in Study 1. Accordingly, we believe that the presentation of the scientists’ activism in Study 1 cannot fully explain the observed effects.

Reviewer 2 Comment 4:

The term “neutral expert” is ambiguous in the context of climate change. For instance, a recent perspective article (van Eck et al., 2024) problematizes the notion of neutrality in climate expertise. The authors should clarify how they operationalize neutrality in their experimental condition and reflect on the normative implications of this framing. <https://www.nature.com/articles/s44168-024-00171-9>

Author Response 4: Thank you for pointing to the lively debate on whether scientists ought to engage in activism. We now include van Eck's work⁵ as follows:

Quote: "Several arguments have been proposed in favour of this notion: (...) that science cannot (and must not) be value-free,⁵" (p. 2)

Quote: "To resolve the activist dilemma described here—balancing the need to create change with the need to maintain a reputation as non-hypocritical and competent—scientists who wish to engage in activism have several options. One option is to separate science and activism activities by highlighting one's role as a private citizen, not an expert, during activism. This approach is generally in line with the pledge to only voice recommendations in line with one's own scientific expertise and in a non-defensive, neutral manner,⁴⁰ but see van Eck et al.⁵" (p. 23)

Reviewer 2 Comment 5:

The theoretical and/or empirical rationale for selecting the many outcome variables and moderators is unclear. For example, why include an 8-item measure of perceived hypocrisy? Under what assumptions would one expect the activist researchers to be perceived as more hypocritical? I encourage the authors to clarify and justify the selection of at least some of the outcome variables and moderators.

Author Response 5: We justify the inclusion of the hypocrisy scale as well as the person evaluation scales as follows:

Quote: "The person evaluation scale covered the most common dimensions of social evaluation;²⁹ we additionally included hypocrisy as it represents a common response to social motive violations and is related to message rejection.³⁰" (p. 8)

Reviewer 2 Comment 6:

While the materials and data are shared on OSF, the study appears not to have been preregistered. This is absolutely fine. However, if this is the case and combined with comment #5, the authors should consider labeling their analyses as exploratory and interpret findings appropriately.

Author Response 6: We agree that the lack of pre-registration is a shortcoming of Study 1, rendering all analyses exploratory (see Quote below). We additionally conducted Study 2 where we pre-registered the observed effects on hypocrisy and competence (expertise). Both hypotheses were confirmed in Study 2 (p. 15).

Quote: "Study 1 suffers from three limitations. First, the study was not pre-registered and no formal predictions were made for these specific effects of competence and hypocrisy to emerge. Accordingly, these findings must be considered exploratory." (p. 13)

Reviewer 2 Comment 7:

Did you include a comprehension check to ensure participants read the target descriptions properly? If not, it may be interesting to control for time spent reading the profiles (e.g., if available in Qualtrics or whatever software was used).

Author Response 7: The condition effects in Study 1 were not moderated by survey completion time (see Quote below). We moreover included reading and attention checks in Study 2. Few participants failed the checks (15 out of 651) and (as pre-registered) we excluded them from the final analysis (p. 15).

Quote: “We observed no significant differences across Topics (Energy saving, Recycling) and no significant Condition × Completion Time interactions (see Supplementary for details).” (p. 11)

Reviewer 2 Comment 8:

The dropout rate (226 incomplete responses) is notable, raising concerns about selective attrition. Can the authors comment on the potential causes and whether attrition was systematically related to any variables of interest?

Author Response 8: To be maximally transparent, we reported all instances of clicks on the study, as opposed to the typical process of reporting only people who started the survey and then dropped out. Accordingly, many cases are reported. We tested if any condition effects emerged on participant attrition, which was not the case. We conclude that systematic attrition likely does not limit the validity of our observed effects. We discuss these analyses in the paper as follows:

Quote Study 1: “Our raw dataset includes 222 incomplete entries and test-runs which are not included in this number; analyses did not show systematic attrition (see Supplementary for details).” (p. 6)

Quote Study 2: “Our raw dataset includes 129 incomplete entries which are not included in this number; analyses did not show systematic attrition (see Supplementary, for details).” (p. 15)

Reviewer 2 Comment 9:

I highly value the power calculations and adjusting for multiple comparisons. However, the blanket statement that “power was sufficient to explore moderators” deserves elaboration, especially given the number of conducted moderation analyses.

Author Response 9: We now include more detail on the power to detect interactions:

Quote Study 1: “The sample was also sufficient to explore moderators yielding a small-to-medium interaction effect of $f = .16$ at $1 - \beta = .95$ or a small effect of $f = .13$ at $1 - \beta = .80$.” (p. 7)

Quote Study 2: “A power analysis using G*power 3.1 (Faul, Erdfelder, Buchner, & Lang, 2009), assuming the previously observed effect of $d = .15$ and setting $1 - \beta = .95$, resulted in a minimum sample size of $N = 580$ for a paired-samples t -test (two-tailed). We aimed to recruit 650 subjects to account for potential drop-outs (see below). The sample was also sufficient to explore moderators yielding a small-to-medium interaction effect of $f = .14$ at $1 - \beta = .95$ or a small effect of $f = .11$ at $1 - \beta = .80$.” (p. 15)

Reviewer 2 Comment 10:

It would be helpful to briefly summarize the descriptive results before discussing the experimental results. For example, the means for competence, morality, and agreement with the comment were relatively high across conditions.

Author Response 10: We now include descriptive results at the onset of the results sections (p. 11 & p. 17, see Table 3). We further discuss the generally positive evaluations of the target scientists as follows:

Quote: “First, across both experimental conditions, evaluations of the scientists were generally positive, indicating that we are observing shifts within an overall favourable impression rather than outright rejection or hostility toward activist scientists. The effects,

therefore, reflect relative differences in warmth and credibility rather than categorical disapproval.” (p. 22)

Reviewer 2 Comment 11:

Were there any differences between the energy-saving and waste-reduction researcher profiles? Given that both were used, this potential source of variation might be interesting to discuss.

Author Response 11: We have conducted counterbalancing checks and now discuss them as follows:

Quote: “We observed no significant differences across Topics (Energy saving, Recycling) and no significant Condition × Completion Time interactions (see Supplementary for details).” (p. 11)

Quote: “We observed no significant differences across Topics (Energy consumption, Consumer products) and entering survey Completion Time as a moderator yielded no significant Condition × Completion Time interactions (see Supplementary for details).” (p. 17)

Reviewer 2 Comment 12:

The abstract is too brief to adequately capture the study's focus and relevance. I encourage you to expand the abstract to provide more clarity on the research design, findings, and implications.

Author Response 12: We have extended abstract as follows:

Quote: “Cost-benefit analyses of whether environmental scientists should engage in activism currently rest on a thin empirical base, despite a lively debate on the topic. There are several potential benefits of scientists’ activism, but some have argued that these benefits might be offset by the potential for activism to undermine public perception of environmental scientists as unbiased and competent. To explore these potential consequences, we asked participants to read two (ostensibly real) profiles of climate scientists that either described themselves as activists or not. Study 1 ($N = 491$) found that a scientist who engaged in conventional activism was seen as slightly less competent and more hypocritical than a scientist who engaged in public science communication, but there was no impact on their persuasiveness. Study 2 ($N = 636$, pre-registered) found that a scientist who engaged in civil disobedience, a more disruptive form of activism, was seen as less competent and more hypocritical than a non-activist scientist who only engaged in teaching and research, with predicted spill-over effects on trust in the scientist’s field. Scientist activists were also downgraded on a range of other dimensions. We draw on these data to guide those who wish to engage in activism without threatening their position as competent, genuine brokers of information.” (p. 1)

Reviewer 2 Comment 13:

The authors use terminology inconsistently throughout the manuscript. I also had to read the method section several times to understand the experiment and measures properly. For example, you refer to the researcher’s “comments” and “messages” somewhat interchangeably, which confused me. Moreover, to me, these comments/messages essentially represent descriptions of researchers without making any clear comments. I worry that this also confused participants; for instance, questions like “What do you think about the person who made the comment?” and “To what extent do you think the person who wrote these comments cares about the environment?” might have been difficult to interpret. The latter obviously cannot be addressed but greater consistency and clarity in referring to the stimuli would be helpful.

Author Response 13: We now consistently refer to “descriptions” of researchers throughout the manuscript. In Study 2, we revised the employed measures accordingly.

Reviewer 2 Comment 14:

Given the concerns outlined above, I believe the discussion of limitations is too superficial. I strongly encourage the authors to expand this section to reflect on limitations related to design, stimuli, sampling, and generalizability. For example, the sample is not representative of the US population, which should be highlighted (perhaps in addition to other quality concerns widely raised about mTurk).

Author Response 14: We substantially extended the discussion of Study 1 limitations (see first Quote below) and designed Study 2 to address these shortcomings (see second Quote below).

Thank you again for serving as a reviewer on our manuscript. We appreciate your input and believe that addressing your comments has further improved our manuscript.

Quote: “Study 1 suffers from three limitations. First, the study was not pre-registered and no formal predictions were made for these specific effects of competence and hypocrisy to emerge. Accordingly, these findings must be considered exploratory. Second, the activism described in the experimental condition was conventional (e.g., attending rallies). However, advocates for scientists’ activism commonly call for more disruptive forms of activism^{5, 6, 25} such as civil disobedience, which is the focus of Study 2. Third, our observed effects were small, potentially due to the materials or the control condition used: The AI-generated background-scenes may have been unconvincing for some participants. Likewise, the description of public engagement with the press in the non-activist scientist condition may have been interpreted as a form of activism by some participants.” (p. 13)

Quote: “The aims of Study 2 were three-fold. First, we sought to provide a confirmatory test of the potential consequences of environmental scientists’ disruptive activism (i.e., civil disobedience). To this end, we pre-registered Study 2 and recruited a sample providing a high power ($1 - \beta = .95$) to detect a small-to-medium effect ($d = .15$). Second, Study 2 adapted the civil disobedience manipulation and measures from past work.²⁵ The consequences of such disruptive activism are somewhat unclear. Social psychological literature in other contexts suggests that more disruptive activism might exacerbate negative consequences for the perception of scientists as impartial brokers of information.³⁶ However, disobedience may also signal a greater urgency of the issue while preserving credibility evaluations, at least in a supportive audience (i.e., among college students).²⁵ Finally, disruptive activists may promote engagement with the cause even if they are personally evaluated negatively.³⁷ Third, Study 2 followed established procedures using text-only materials and employed a control condition only describing research and teaching activities related to environmental issues. Because demographics and political preferences may be related to environmental- and activism-related attitudes,²³ we recruited a quota-based sample of Americans, stratified by age, gender, and political orientation to approximate national distributions on these variables.” (p. 14)

"Navigating the Credibility Risks of Environmental Scientists' Activism"

Response to Reviewer 1 Comments

Reviewer 1 Comment 1:

I thank the authors for carefully addressing my points. I am impressed by the fact that the authors conducted an additional, pre-registered study to further substantiate and validate their initial findings. I agree with the other reviewer that in the initial submission it felt like the authors had already made up their minds about whether climate-related researchers should engage in activism. I found the revised version to be much more balanced and the limitations described more clearly and transparently. I have no further comments, thank the authors for their careful and substantial revision, and recommend publication.

Author Response 1: Thank you for serving as a reviewer on our revised manuscript and your constructive comments. We appreciate your endorsement for publication.

Response to Reviewer 2 Comments

Reviewer 2 Comment 1:

Thank you for carefully considering my comments. The manuscript has improved substantially, especially with the more extended introduction and addition of Study 2. I therefore only have a few minor comments:

1. I appreciate the authors' inclusion of descriptive results in Studies 1 and 2. However, I encourage adding a few sentences interpreting those results, such as commenting on the mean levels of competence, hypocrisy, morality, etc. This will help contextualize the experimental results. For example, is it a cause of concern that perceived hypocrisy slightly increases for activists versus non-activists if the mean perceived hypocrisy is still quite low?

Author Response 1: We have added an interpretation of the descriptive results to each study results (first and second quote below) and the General Discussion (third quote below).

Quote: "Evaluations of scientists were generally positive, as indicated by relatively high ratings of competence, high attributions of concern for the environment, and low hypocrisy scores across the conditions." (p. 11)

Quote: "Evaluations of scientists were again generally positive, with trait ratings and attributions of concern for the environment above the mid-point in both conditions." (p. 18)

Quote: "Overall, scientists were evaluated quite positively, as indicated by high average ratings of competence, morality, and trust as well as low hypocrisy (Table 3)." (p. 22)

Reviewer 2 Comment 2:

2. Out of curiosity, what is the expected or potential causal mechanism for this finding in Study 2 – "Finally, scientists who engaged (vs. did not engage) in activism led participants to report a lower intention to act pro-environmentally."?

Author Response 2: We speculate about the underlying mechanisms as follows:

Quote: “Interestingly, scientists’ activism also lowered participants’ own stated intentions to act pro-environmentally. It is possible that disruptive activism triggered perceptions of norm violation and bias, which reduced trust not only in the individual scientists but in their broader scientific agenda. These credibility losses could have led people to distance themselves from the cause and to morally dissociate from the behaviour being advocated.” (p. 21)

Reviewer 2 Comment 3:

3. This statement in the general discussion reflects a theory of change that, while logical, may not be universally true for achieving political change: “In both studies these effects were particularly pronounced among those members of the community who need to be reached to have an impact: those with relatively low trust in science and low levels of pro-environmental behaviour.” In the US, political change often happens without public support (e.g., anti-abortion legislation), and environmentally significant behavior change may occur through changes in economic incentives, infrastructure, etc., without necessarily changing people’s “minds” first.

Author Response 3: Thank you for this insightful and constructive comment. In response, we have expanded our discussion of the positive consequences of disruptive activism in the General Discussion to reflect this broader perspective (see Quote below). Specifically, we now engage with the work of Tufekci (2017), who argues that disruptive protest can function as a mechanism for building movement capacity—across narrative, disruptive, and institutional dimensions—regardless of whether it garners widespread public support. We also incorporate Han and Barnett-Loro’s (2018) discussion of how movement organizations develop strategic capacity beyond shifting individual attitudes.

These additions aim to briefly clarify that while our study focuses on the psychological consequences of observing scientists engage in activism, this focus does not assume that public support is the sole or even primary goal of all disruptive protest actions. Rather, we position our findings as relevant within a broader strategic landscape, where scientists must navigate the potential trade-offs between immediate public opinion and the potential for long-term capacity building. We hope this revision addresses your concern and strengthens our contribution.

Quote: “Second, public approval is not the sole metric of effective activism. Those engaging in activism may, in fact, not aim for approval but seek disruption to signal power,³⁹ build momentum,⁴⁰ or attract media attention.⁴¹ The assumption is that the resulting tension is required for societal change—even if it comes at the cost of alienating some members of the public. According to this view, disruptive and divisive actions may nonetheless be consequential if they attract media attention, shift public agendas, or pressure decision-makers to act (see, e.g., Ostarek, et al.⁴².” (p. 23)

Reviewer 2 Comment 4:

4. In the general discussion, I felt there was a lack of reflection on the study context. To what extent do the results and their implications generalize to other contexts than the United States, given its unique political system and political climate?

Author Response 4: We now reflect on the study context as indicated in the quote below. Thank you again for your constructive comments and your support in getting this manuscript ready for publication.

Quote: “Both reported studies were conducted in the US, a context highly polarized on science and environmental issues. For instance, Cologna et al.²² observed that while average endorsement of scientists’ public engagement was moderate in the US, these ratings were strongly correlated with participants’ reported trust in science. This relationship between trust in science and the endorsement of scientists’ engagement was weaker

or even reversed in other countries. It is thus possible that the observed effects would be attenuated in contexts where science and the environment are less polarizing topics. Accordingly, future research should assess how engaging in activism influences the perception of scientists across cultures.” (p. 24)

Additional Author Response: We spotted an ambiguity in our description of Dablander et al.’s (2025) study. In their paper, the authors stated that “To assess the credibility of the scientist engaging in action, we adapted an item from Friedman [31] (...) **Note that this item was only presented in the scientist engagement conditions that include Dr Fraser, thus not in the control condition.**” (p. 6, emphasis added) Accordingly, we rephrased our description as follows:

Quote: “A recent, high-powered registered report used vignettes to investigate the influence of scientists’ participation in protests against oil and gas drilling in an experimental design. Vignettes either did not mention scientists (control condition), mentioned a scientist’s endorsement of protests, or mentioned a scientist’s active participation. The study found no evidence that scientists’ participation in activism undermined their credibility or public trust in environmental science as compared **to only endorsing the protests (this variable was not assessed in the control condition).**” (p. 4, emphasis added)